# Dynamic Multi-Task Weight Adaptation for Efficient Sentiment Analysis Fine-Tuning on LLMs

## Abstract

Sentiment analysis is crucial across domains from business intelligence to financial forecasting, with large language models (LLMs) emerging as powerful tools for financial text analysis. However, fine-tuned LLMs often exhibit suboptimal performance due to severe data distribution imbalance in financial datasets, where neutral sentiments dominate while extreme sentiments remain underrepresented, causing strong bias toward over-represented regions and poor accuracy for critical extreme sentiment values. To address this limitation, we propose a novel multi-task learning framework that incorporates both regression and classification objectives, along with data-aware stratification (DAS) algorithm and dynamic weight adapter (DWA) module. The multi-task learning design introduces auxiliary classification tasks to assist sentiment polarity analysis, providing complementary supervision that helps models better understand sentiment boundaries. The DAS algorithm mitigates data distribution imbalance through automatic threshold optimization, creating balanced categorical mapping for the classification task. The DWA module dynamically adjusts task weights based on gradient information and batch characteristics during training, addressing the varying task complexities and convergence rates inherent in multi-task optimization. Our approach decomposes the data distribution imbalance problem into two manageable sub-problems: balanced categorical mapping and adaptive task weighting. Comprehensive experiments using different model configurations demonstrate superior performance. Our framework achieves an average improvement of 12.36% in Mean Squared Error (MSE) and 1.41% in Accuracy (ACC) across multiple datasets compared with previous work.

## 1 Introduction

Sentiment analysis has become an essential tool for businesses to understand customer opinions and feedback, with applications ranging from supply chain management (Goel et al., 2024) to demand forecasting (Bi et al., 2021) and analyzing product reviews (Chacón-Cardona et al., 2024). In the financial sector, sentiment analysis is particularly crucial for forecasting stock market trends (Balasudarsun et al., 2022), predicting cryptocurrency prices (Bouteska et al., 2024), and forecasting exchange rates (Ding et al., 2024).

However, when fine-tuning LLMs for financial sentiment polarity analysis, we observed significant performance degradation due to severe data distribution imbalance. Specifically, financial datasets typically exhibit heavy concentration in the neutral range while extreme sentiments (strong positive and strong negative) remain underrepresented (Omarkhan et al., 2021; Davidovic & McCleary, 2025), causing models to develop strong bias toward over-represented regions and poor prediction accuracy for critical extreme sentiment values (Mujahid et al., 2024; Nassr et al., 2025).

Traditional approaches to address data distribution imbalance include data augmentation techniques (Assunção et al., 2024), cost-sensitive learning methods (Kouzani et al., 2024), and threshold-moving strategies (Salekshahrezaee et al., 2023). However, these static solutions suffer from fundamental limitations: data augmentation may introduce artificial patterns that deviate from authentic financial discourse (Schaudt et al., 2023), cost-sensitive methods require extensive hyperparame-

Figure 1: Comparison between single-task approach (**A**) and our proposed multi-task learning approach (**B**). Our approach introduces auxiliary classification tasks to complement sentiment regression, DAS algorithm for balanced data mapping, and DWA module for adaptive task weighting, mitigating the impact of imbalanced datasets and achieving performance improvements.

ter tuning and often lack sufficient adaptability (Khan et al., 2017; Edsa et al., 2025), and threshold adjustment approaches fail to address learning bias during training (Johnson & Khoshgoftaar, 2021).

To address these challenges, we noval propose a multi-task learning framework that transforms the single-task regression problem into complementary regression and classification objectives, as shown in Figure 1. The classification task provides auxiliary supervision (Yao et al., 2023) that helps models better understand sentiment boundaries and reduces neutral sample dominance (Hong et al., 2024). While multi-task learning offers these advantages, it also introduces complexity through varying task difficulties and convergence rates (Senushkin et al., 2023), which can lead to suboptimal performance with fixed weighting (Wang et al., 2021).

Our framework decomposes the challenging data distribution imbalance problem into two manageable sub-problems: introducing auxiliary classification tasks to complement the regression objective and dynamic adaptive task weight module to handle inter-task difficulty discrepancies. Specifically, we introduce a DWA module that dynamically adjusts task weights based on gradient information and batch characteristics during training, integrated with a DAS algorithm for automatic sentiment threshold determination and LoRA for parameter-efficient fine-tuning (PEFT). By solving these two simpler problems simultaneously, we effectively address the overall challenge while maintaining computational efficiency. The main contributions of this paper are as follows:

- We identify performance gaps in LLMs fine-tuned for sentiment polarity analysis within the financial domain: (1) severe sentiment distribution imbalance biases models toward over-represented neutral regions during training, and (2) constant-weight multi-task learning approaches fail to accommodate varying task difficulties between regression and classification objectives.

- We propose a novel multi-task learning framework that incorporates auxiliary classification tasks to complement sentiment regression and a DAS algorithm for balanced categorical mapping, effectively addressing imbalanced data distribution challenges.

- We introduce a plug-and-play DWA module that dynamically adjusts task weights based on gradient information and batch characteristics during training, adaptively handling varying task difficulties.

- Our framework outperforms baseline approaches across various datasets, achieving average improvements of 12.36% in MSE and 1.41% in accuracy compared with previous work.

## 2 MOTIVATION

In this section, we introduce data-level and task-level challenges in LLM fine-tuning for financial sentiment analysis.

**Data-level Challenge: Imbalanced Data Distribution.** In sentiment polarity analysis tasks for financial texts, we observe that LLM performance is unsatisfactory after fine-tuning due to severe data distribution imbalance. To investigate this limitation, we analyze the data distribution patterns shown in Figure 2. Our analysis reveals that training sets typically suffer from significant data imbalance, with excessive concentration in the neutral sentiment range. When fine-tuned models make predictions on test sets, they demonstrate pronounced bias toward sentiment ranges that are over-represented during training (Krawczyk, 2016). The comparative analysis in Figure 2(D) shows substantial distributional discrepancies. More details can be found in Appendix A.1.

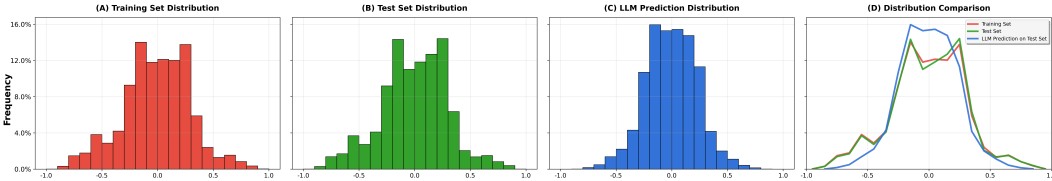

Figure 2: Data distribution analysis revealing imbalance challenges in sentiment analysis, including: (A) Training set distribution, (B) Test set distribution, (C) Fine-tuned LLM predictions on test set, and (D) Comparative analysis.

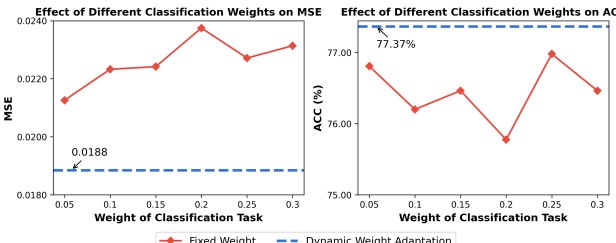

Figure 3: Effect of different task weights on MSE and ACC in multi-task learning, compared with the performance of our proposed framework incorporating the DWA module.

**Task-level Challenge: Inter-task Difficulty Discrepancy.** To mitigate data imbalance issues, we incorporate a classification task alongside the regression task to better capture sentiment tendencies embedded in the text. However, in our multi-task learning experiments, we observe that the model's performance varies across different tasks as a result of varying loss function weights, as shown in Figure 3. As the weight of the classification task increases, the MSE increases while the ACC decreases, indicating a fundamental trade-off between the regression and classification objectives. The optimal balance appears to be around a classification weight of 0.25, where MSE is low and ACC is relatively high. Upon integration of the DWA module, the model achieves superior performance across all metrics, yielding the lowest MSE of 0.0188 and the highest ACC of 77.37%. This sensitivity to weight configuration reveals the fundamental inter-task difficulty discrepancy that motivates our dynamic weighting approach. More details can be found in Appendix A.2.

## 3 METHOD

In this section, we introduce our multi-task learning framework, DAS algorithm and our framework incorporating the DWA module with LoRA.

### 3.1 MULTI-TASK LEARNING FRAMEWORK

Our multi-task learning framework simultaneously performs regression and classification tasks for enhanced sentiment analysis. We employ LLMs as the backbone to generate text embeddings, which are fed into task-specific heads. As shown in Figure 4(A), the pre-trained LLM processes input text through multiple Transformer blocks. The framework incorporates two parallel task modules: a regression module (Figure 4(B)) that predicts continuous sentiment polarity scores using MSE loss $L_r$, and a classification module (Figure 4(C)) that maps embeddings to discrete sentiment categories using cross-entropy loss $L_c$. To jointly optimize both tasks, we define the multi-task learning loss as $L_{\mathrm{mtl}} = w_r L_r + w_c L_c$, where $w_r$ and $w_c$ are task weights during training that manage the trade-off between tasks. More details can be found in Appendix A.3.

### 3.2 DATA-AWARE STRATIFICATION ALGORITHM

To migrate imbalanced data distribution issues caused by manual sentiment label division, we propose a DAS algorithm that constructs a balanced classification task by threshold selection to map sentiment scores to categories. The dataset is defined as $D = \{(x_i, y_i, z_i)\}_{i=1}^{N}$, where $N$ is the total number of samples, $x_i$ denotes text, $y_i \in [-1, 1]$ corresponds to continuous sentiment polarity scores, and $z_i \in \{z_1, z_2, \ldots, z_K\}$ represents sentiment category labels for $K$ categories.

Figure 4: This framework illustrates LLM-based sentiment analysis with three components: (A) LLMs as the backbone combined with a task-specific module, (B) a regression task module, and (C) a classification task module that demonstrates the multi-task framework handling both regression and classification tasks simultaneously. More details can be found in Appendix A.3.

**Threshold Selection.** We formulate the threshold selection as an optimization problem that minimizes within-class variance while preserving distributional balance (Wu et al., 2023; Nguyen et al., 2022). For a threshold set $\mathcal{T} = \{\tau_0, \tau_1, \ldots, \tau_{K-1}\}$ that divides the sentiment space into $K$ discrete categories, we define the objective function as:

$$\mathcal{L}(\mathcal{T}) = \sum_{k=1}^{K} \frac{|S_k|}{N} \cdot \text{Var}(S_k) + \lambda \sum_{k=1}^{K-1} (\tau_k - \tau_{k-1} - \Delta_{\text{target}})^2, \tag{1}$$

where $S_k = \{y_i : y_i \in I_k(\mathcal{T})\}$ denotes the set of sentiment scores within the $k$-th interval $I_k(\mathcal{T})$, $|S_k|$ represents the cardinality of set $S_k$, $\text{Var}(S_k)$ is the variance within stratum $k$, $\lambda$ is the regularization parameter balancing variance minimization and uniform spacing, and $\Delta_{\text{target}} = \frac{2}{K-1}$ ensures uniform interval spacing. More details can be found in Appendix A.4.

**Threshold Initialization.** To ensure balanced representation across sentiment categories, we initialize thresholds using adaptive quantiles (Zhang et al., 2023; Gao et al., 2025):

$$\tau_k^{(0)} = Q_{\alpha_k}(\{y_i\}_{i=1}^{N}), \quad \text{where } \alpha_k = a + b \cdot \frac{k}{K-1}, \tag{2}$$

where $Q_{\alpha_k}$ represents the $\alpha_k$-quantile operator, and $a$ and $b$ are parameters that define the quantile range for threshold initialization, ensuring balanced representation across sentiment categories while preserving extreme sentiment samples. More details are provided in Appendix A.4.

**Threshold Update.** We optimize the threshold set using gradient descent:

$$\tau_k^{(t+1)} = \tau_k^{(t)} - \eta \frac{\partial \mathcal{L}(\mathcal{T})}{\partial \tau_k}, \tag{3}$$

subject to the ordering constraint $\tau_{k-1} < \tau_k < \tau_{k+1}$, where $\eta$ is the learning rate. The algorithm converges when $\|\mathcal{T}^{(t+1)} - \mathcal{T}^{(t)}\|_2 < \epsilon$, where $\epsilon$ is the convergence threshold.

**Mapping Function.** The optimal mapping function $f^* : [-1, 1] \rightarrow \{z_1, \ldots, z_K\}$ is constructed as $f^*(y) = z_k$ if $y \in I_k(\mathcal{T}^*)$, where the intervals are defined as:

$$I_k(\mathcal{T}^*) = \begin{cases} (-1, \tau_0^*], & \text{if } k = 1 \\ (\tau_{k-1}^*, \tau_k^*], & \text{if } k = 2, \ldots, K-1 \\ (\tau_{K-1}^*, 1], & \text{if } k = K \end{cases} . \tag{4}$$

This algorithm provides balanced discrete labels $\{z_i\}_{i=1}^{N}$ through optimal thresholds that integrate with our DWA module to enhance the multi-task framework performance.

## 3.3 DYNAMIC WEIGHT ADAPTER

To address these challenges mentioned in motivation, we propose a DWA module as shown in Figure 5(A), a learnable neural network that automatically balances task contributions and mitigates data imbalance issues during training. Our approach addresses challenges of inter-task difficulty discrepancy and imbalanced data distribution.

**Solution #1: Inter-task Difficulty Discrepancy.** Due to the different magnitudes of the loss functions for regression and classification tasks, directly summing them may cause one task to dominate

Figure 5: Multi-task learning framework architecture. (A) Framework workflow with DWA integration, (B) DWA module internal structure, and (C) LoRA implementation in transformer layers.

the entire training process. To balance the contributions of different tasks, we introduce a gradient-based weighting method. Specifically, we define the total multi-task loss function $L_{\mathrm{mtl}}^t$ as follows:

$$L_{\mathrm{mtl}}^t = \lambda_r^t w_r^t L_r^t + \lambda_c^t w_c^t L_c^t, \tag{5}$$

where $L_r^t$ and $L_c^t$ represent the losses of the regression and classification tasks at step $t$, respectively, $w_r^t$ and $w_c^t$ are the task-specific weights, and $\lambda_r^t$ and $\lambda_c^t$ are the gradient-based weighting coefficients. At each training step, we compute gradient-based weighting coefficients that inversely weight tasks based on their gradient magnitudes:

$$\lambda_r^t = \frac{|\nabla L_c^t|}{|\nabla L_r^t| + |\nabla L_c^t|}, \quad \lambda_c^t = \frac{|\nabla L_r^t|}{|\nabla L_r^t| + |\nabla L_c^t|}. \tag{6}$$

This inverse weighting mechanism ensures that tasks producing larger gradients receive proportionally smaller coefficients, thereby balancing the learning progress and scaling both tasks to similar magnitudes in $L_{\mathrm{mtl}}^t$.

**Solution #2: Imbalanced Data Distribution.** We introduce a regularization term and class-specific weights into the classification loss function. The regularization term $-\alpha \log p_k^t$ encourages the model to pay attention to minority classes with smaller proportions, while the class-specific weights $v_k^t$ are inversely proportional to the proportions $p_k^t$:

$$v_k^t = \frac{1}{(p_k^t)^\beta}, \tag{7}$$

where $\beta$ is a parameter that controls the class balancing.

We use $D^t$ to represent the batch of data sampled at time step $t$. For each batch $D^t$, we calculate the proportion of samples $p_k^t$ belonging to class $k$ as:

$$p_k^t = \frac{|D_k^t|}{|D^t|}, \tag{8}$$

where $|D_k^t|$ and $|D^t|$ denote the number of samples in batch $D^t$ belonging to class $k$ and the total number of samples in batch $D^t$, respectively.

To incorporate the regularization term $-\alpha \log p_k^t$ and the class-specific weights $v_k^t$ into the classification loss function, we define the imbalanced data distribution loss $L_{\mathrm{imb}}^t$ conditioned on $D^t$ as:

$$L_{\mathrm{imb}}^t(D^t) = \sum_{k=1}^B v_k^t (p_k^t L_{c,k}^t - \alpha \log p_k^t). \tag{9}$$

This loss captures the sample distribution and class proportions specific to the current batch, allowing the model to dynamically adapt to the characteristics of each batch during training. We then incorporate the imbalanced data distribution loss $L_{\mathrm{imb}}^t(D^t)$ into the total multi-task loss function $L_{\mathrm{mtl}}^t$ established for addressing inter-task difficulty discrepancy, along with the task-specific weights $w_r^t$ and $w_c^t$:

$$L_{\mathrm{mtl}}^t(D^t) = \lambda_r^t w_r^t L_r^t + \lambda_c^t w_c^t L_{\mathrm{imb}}^t(D^t) = \lambda_r^t w_r^t L_r^t + \lambda_c^t w_c^t \sum_{k=1}^B v_k^t (p_k^t L_{c,k}^t - \alpha \log p_k^t), \tag{10}$$

where $\lambda_r^t$ and $\lambda_c^t$ are the gradient-based weighting coefficients, $v_k^t$ are the class-specific weights, and $L_{c,k}^t$ is the classification loss for class $k$ at time step $t$. The total multi-task loss $L_{\mathrm{mtl}}^t$ is conditioned on the current batch $D^t$, allowing the model to adapt to the specific characteristics of $D^t$.

To learn the hyperparameters $\alpha$ and $\beta$, we treat them as learnable parameters of the DWA module and update them using gradient descent along with the other parameters of the module. During

the backward pass, the gradients of the DWA module loss $L_{\text{mtl}}^t(D^t)$ with respect to $\alpha$ and $\beta$ are computed as follows:

$$\frac{\partial L_{\text{mtl}}^t(D^t)}{\partial \alpha} = -\lambda_c^t w_c^t \sum_{k=1}^{B} v_k^t \log p_k^t. \tag{11}$$

To compute the gradient with respect to $\beta$, we apply chain rule and substitute the expression for $\frac{\partial v_k^t}{\partial \beta}$:

$$L_{\text{mtl}}^t(D^t) = \lambda_r^t w_r^t L_r^t + \lambda_c^t w_c^t L_{\text{imb}}^t(D^t) = \lambda_r^t w_r^t L_r^t + \lambda_c^t w_c^t \sum_{k=1}^{B} v_k^t (p_k^t L_{c,k}^t - \alpha \log p_k^t). \tag{12}$$

The hyperparameters $\alpha$ and $\beta$ are then updated using an optimizer such as Adam or AdamW.

To obtain the task-specific weights $w_r^t$ and $w_c^t$, we pass the gradient-weighted regression loss $\lambda_r^t L_r^t$, the gradient-weighted imbalanced data distribution loss $\lambda_c^t L_{\text{imb}}^t(D^t)$, and the current batch $D^t$ through the DWA module as illustrated in Figure 5(B):

$$w_r^t, w_c^t = \text{DWA}(\lambda_r^t L_r^t, \lambda_c^t L_{\text{imb}}^t(D^t), D^t). \tag{13}$$

Specifically, we first concatenate these weighted losses and map them to a hidden space through a fully connected layer (FC$_1$) followed by a ReLU activation function:

$$\mathbf{h}^t = \text{ReLU}(\text{FC}_1([\lambda_r^t L_r^t, \lambda_c^t L_{\text{imb}}^t(D^t)])) = \text{ReLU}(\text{FC}_1([\lambda_r^t L_r^t, \lambda_c^t \sum_{k=1}^{B} v_k^t (p_k^t L_{c,k}^t - \alpha \log p_k^t)])).$$
$$\tag{14}$$

The hidden representation $\mathbf{h}^t$ encodes the information from the weighted losses and the current batch data. Next, we use another fully connected layer (FC$_2$) to map the hidden representation $\mathbf{h}^t$ to a two-dimensional vector $\mathbf{s}^t$: $\mathbf{s}^t = \text{FC}_2(\mathbf{h}^t)$. Finally, we pass the vector $\mathbf{s}^t$ through a Softmax function to obtain the task-specific weights $w_r^t$ and $w_c^t$ based on $D^t$:

$$[w_r^t, w_c^t] = \text{Softmax}(\mathbf{s}^t). \tag{15}$$

In summary, the DWA module receives loss information and outputs adaptive task weights. The gradient-based coefficients $\lambda_r^t$ and $\lambda_c^t$ address inter-task difficulty discrepancy, while the learnable parameters $\alpha$ and $\beta$ handle imbalanced data distribution, enabling the model to adapt to varying batch characteristics and optimize multi-task performance.

## 3.4 LoRA Integration with Multi-task Framework

We implement LoRA (Hu et al., 2021) for PEFT of LLMs. LoRA injects trainable low-rank decomposition matrices into each layer, reducing parameters while maintaining performance.

LoRA introduces rank-$r$ matrices $U \in \mathbb{R}^{d \times r}$ and $V \in \mathbb{R}^{r \times d}$ into each attention block and feed-forward network, as shown in Figure 5(C). Our multi-task framework integrates LoRA into the LLM backbone for PEFT. The figure illustrates how LoRA is integrated into the Transformer, with pretrained weights $W$ remaining frozen while low-rank matrices are trained. The adjusted weight matrix $W'$ during forward propagation is:

$$f(x) = (W + UV) \cdot X + b, \tag{16}$$

where $b$ is the bias, and $X$ is the input as illustrated in Figure 5(C). Only $U$ and $V$ are updated, significantly reducing trainable parameters and potentially accelerating training.

## 4 Experiments

### 4.1 Implementation Details

**Software and Hardware.** We use Ubuntu 22.04, Python 3.9.19, PyTorch 2.3.1, PEFT 0.12.0, and CUDA 12.4, running on a system with 32GB RAM and an NVIDIA RTX 3090Ti GPU with 24GB VRAM.

Table 1: Benchmark models and experimental settings.

| Benchmark | Model | Task |
|---|---|---|
| B1 | RoBERTa-Large (Liu et al., 2019) | Regression |
| B2 | Twitter-RoBERTa-Large (Loureiro et al., 2023) | Regression |
| B3 | Twitter-RoBERTa-Large (Loureiro et al., 2023) | Multi-task (Constant Weight) |
| B4-DWA | Twitter-RoBERTa-Large (Loureiro et al., 2023) | Multi-task (Dynamic Weight) |
| B5-DWA | TinyLlama-1.1B (Zhang et al., 2024)7 | Multi-task (Dynamic Weight) |
| B6-DWA | Qwen2-0.5B (Yang et al., 2024) | Multi-task (Dynamic Weight) |
| B7-DWA+LoRA | Twitter-RoBERTa-Large (Loureiro et al., 2023) | Multi-task (Dynamic Weight) |
| B8-DWA+LoRA | TinyLlama-1.1B (Zhang et al., 2024) | Multi-task (Dynamic Weight) |
| B9-DWA+LoRA | Qwen2-0.5B (Yang et al., 2024) | Multi-task (Dynamic Weight) |

Table 2: Performance comparison between constant weight and DWA approaches.

| Model | Dataset | Multi-Task Learning with Constant Weight | | | | | | | Multi-Task Learning with DWA | | | | | | |
|---|---|---|---|---|---|---|---|---|---|---|---|---|---|---|---|
| | | Epoch* | MSE | ACC | Prec. | F1 | GPU(MiB) | Time(s) | Epoch* | MSE | ACC | Prec. | F1 | GPU(MiB) | Time(s) |
| Twitter RoBERTa-Large | NEU | 72 | 0.0231 | 75.89 | 75.84 | 75.76 | 21,892 | 71,594 | 66 | 0.0192 | 76.89 | 77.78 | 76.98 | 22,390 | 116,093 |
| | FXE | 63 | 0.0244 | 75.12 | 75.23 | 75.18 | 21,761 | 58,577 | 58 | 0.0205 | 76.45 | 77.12 | 76.78 | 22,300 | 94,985 |
| | INV | 91 | 0.0222 | 76.56 | 76.49 | 76.41 | 21,783 | 130,171 | 93 | 0.0187 | 77.69 | 78.74 | 78.02 | 22,345 | 211,078 |
| Qwen 0.5 | NEU | 65 | 0.0247 | 74.89 | 75.12 | 74.78 | 22,564 | 75,738 | 58 | 0.0220 | 76.23 | 77.34 | 76.31 | 22,406 | 122,998 |
| | FXE | 56 | 0.0252 | 74.67 | 74.89 | 74.56 | 22,451 | 61,719 | 60 | 0.0231 | 75.45 | 76.45 | 75.34 | 22,474 | 99,676 |
| | INV | 89 | 0.0241 | 75.18 | 75.41 | 75.02 | 22,519 | 138,671 | 87 | 0.0213 | 76.45 | 77.68 | 76.54 | 22,519 | 224,231 |
| TinyLlama 1.1B | NEU | 76 | 0.0204 | 76.89 | 77.12 | 76.56 | 21,128 | 89,621 | 69 | 0.0182 | 77.78 | 78.12 | 77.56 | 21,699 | 145,365 |
| | FXE | 67 | 0.0215 | 76.23 | 76.78 | 76.12 | 21,107 | 73,767 | 62 | 0.0192 | 77.34 | 77.89 | 77.23 | 21,721 | 119,282 |
| | INV | 93 | 0.0196 | 77.34 | 77.56 | 77.01 | 21,023 | 160,384 | 96 | 0.0177 | 78.12 | 78.45 | 77.89 | 21,634 | 259,340 |

**Benchmark.** We use different benchmarks to verify our framework as shown in Table 1. We establish three baseline benchmarks (B1-B3) and implement our proposed DWA framework across six configurations: three with DWA only (B4-B6) and three with DWA+LoRA integration (B7-B9).

**Evaluation Metrics.** For the regression task, we utilize MSE, Mean Absolute Error (MAE), Root Mean Squared Error (RMSE), and the Coefficient of Determination ($R^2$). For the classification task, we use ACC, weighted Precision (Prec.), and weighted F1 score. We also measure GPU(MiB) (GPU memory consumption), Time (total training time in seconds to complete 100 epochs), and Time Usage (relative training time compared to full fine-tuning, where $<100\%$ indicates acceleration). More details can be found in Appendix A.6.

**Dataset.** We use three financial text datasets: NEU, FXE, and INV (More details about the datasets can be found in Appendix A.7.), containing news and analysis on EUR/USD exchange rates. These datasets are randomly split into training and validation sets with a 9:1 ratio.

**Hyperparameters.** Batch sizes are adjusted from 5 to 10 based on memory limitations. Input texts exceeding 512 tokens are truncated to the first 512 tokens, with training incorporating 100 warmup steps and cosine decay for learning rate scheduling. Random seeds are set to 42 for Python, NumPy, and PyTorch to ensure reproducibility. For DAS algorithm, the final labels are set to 5 categories. The DWA module uses Adam optimizer with a learning rate of 0.001. LoRA configurations employ rank values of 128, 256, 384, and 512, with alpha values equal to the rank and dropout rate of 0.05 to prevent overfitting. More details can be found in Appendix A.8.

## 4.2 MAIN RESULTS

**DWA vs Constant Weight Performance.** Table 2 presents the performance comparison between constant-weight multi-task learning and our proposed DWA framework across three model architectures. The DWA approach consistently outperforms the constant-weight baseline across all configurations. For Twitter-RoBERTa-Large, DWA achieves MSE reductions of 16.1-16.8% and accuracy improvements of 1.0-1.3%. Improvements are observed for Qwen 0.5B and TinyLlama 1.1B, with consistent gains in all metrics and faster convergence. More details can be found in Appendix A.9.

Across all model architectures and datasets, our DWA framework achieves an average improvement of 12.36% in MSE and 1.41% in ACC compared to constant-weight approaches. These results provide strong evidence for the effectiveness of our DWA module. The consistent improvements across different model scales and datasets demonstrate that the DWA mechanism adapts to varying task complexities and data distributions, validating our approach to handling inter-task difficulty discrepancy and imbalanced data distribution challenges.

**Dynamic Weight Evolution.** Figure 6 illustrates the changes in the dynamic adaptive loss function during training, with the weights of the classification and regression tasks set to 1. The upper plot shows the dynamic adjustment of task weights during training. The initial weight of the classification task appears at approximately 0.5 and exhibits an oscillating decrease, stabilizing around 0.15 after approximately 1600 batches. The lower plot displays the average weights of the classification and regression tasks throughout the entire training process. The average weight of the classification task gradually decreases from about 0.3 to 0.1 over the first 25 epochs and continues to fluctuate

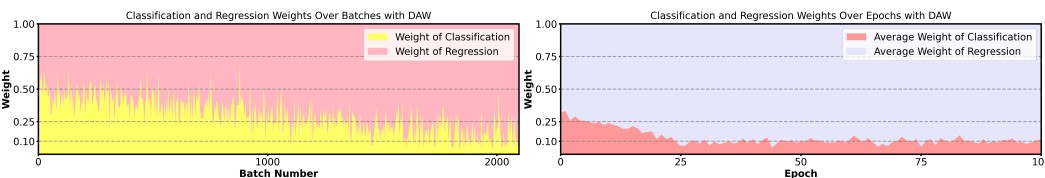

Figure 6: DWA module adjusts task weights during training.

Table 3: LoRA integration with our DWA framework: performance across different ranks.

| Model | Dataset | LoRA (Rank = 128) | | | | | | | | LoRA (Rank = 256) | | | | | | | |
|---|---|---|---|---|---|---|---|---|---|---|---|---|---|---|---|---|---|
| | | Epoch* | MSE | ACC | Prec. | F1 | GPU(MiB) | Time Usage | Batch | Epoch* | MSE | ACC | Prec. | F1 | GPU(MiB) | Time Usage | Batch |
| Twitter RoBERTa-Large | NEU | 66 | 0.0193 | 76.67 | 77.56 | 76.78 | 22,875 | 107.41% | 10 | 50 | 0.0192 | 76.95 | 77.95 | 77.05 | 24,151 | 87.77% | 10 |
| | FXE | 72 | 0.0206 | 76.23 | 76.89 | 76.34 | 22,738 | 133.97% | 10 | 58 | 0.0204 | 76.52 | 77.25 | 76.85 | 24,224 | 116.24% | 10 |
| | INV | 85 | 0.0189 | 76.78 | 78.34 | 77.12 | 22,852 | 109.50% | 10 | 64 | 0.0188 | 77.75 | 78.80 | 78.10 | 24,224 | 88.85% | 10 |
| Qwen 0.5 | NEU | 58 | 0.0222 | 75.89 | 76.67 | 75.98 | 23,415 | 106.03% | 8 | 43 | 0.0218 | 76.30 | 77.45 | 76.40 | 23,825 | 85.26% | 8 |
| | FXE | 65 | 0.0232 | 75.23 | 76.23 | 75.12 | 23,328 | 139.80% | 8 | 52 | 0.0230 | 75.52 | 76.55 | 75.45 | 23,892 | 120.27% | 8 |
| | INV | 76 | 0.0215 | 76.12 | 77.23 | 76.34 | 23,376 | 101.39% | 8 | 58 | 0.0212 | 76.52 | 77.75 | 76.65 | 23,856 | 83.44% | 8 |
| TinyLlama 1.1B | NEU | 69 | 0.0183 | 77.56 | 77.89 | 77.34 | 22,198 | 117.45% | 7 | 52 | 0.0182 | 77.85 | 78.20 | 77.65 | 23,341 | 95.57% | 6 |
| | FXE | 76 | 0.0194 | 77.01 | 77.67 | 77.12 | 22,154 | 146.28% | 7 | 62 | 0.0192 | 77.42 | 77.95 | 77.35 | 23,387 | 128.87% | 6 |
| | INV | 89 | 0.0179 | 77.89 | 78.23 | 77.78 | 22,201 | 125.98% | 7 | 67 | 0.0177 | 78.20 | 78.50 | 78.05 | 23,423 | 102.17% | 6 |
| **Model** | **Dataset** | LoRA (Rank = 384) | | | | | | | | LoRA (Rank = 512) | | | | | | | |
| Twitter RoBERTa-Large | NEU | 30 | 0.0191 | 77.03 | 78.15 | 77.18 | 19,609 | 56.61% | 7 | 18 | 0.0190 | 77.12 | 78.03 | 77.29 | 22,403 | 38.38% | 8 |
| | FXE | 35 | 0.0203 | 76.67 | 77.38 | 76.95 | 19,628 | 75.51% | 7 | 21 | 0.0202 | 76.73 | 77.29 | 77.08 | 22,403 | 51.22% | 8 |
| | INV | 39 | 0.0184 | 77.89 | 78.95 | 78.23 | 19,550 | 58.23% | 7 | 23 | 0.0183 | 78.07 | 79.15 | 78.38 | 22,336 | 38.80% | 8 |
| Qwen 0.5 | NEU | 25 | 0.0218 | 76.41 | 77.58 | 76.62 | 20,128 | 76.35% | 6 | 14 | 0.0217 | 76.58 | 77.85 | 76.79 | 22,894 | 47.86% | 7 |
| | FXE | 29 | 0.0229 | 75.67 | 76.72 | 75.73 | 20,185 | 102.94% | 6 | 17 | 0.0228 | 75.83 | 76.95 | 75.91 | 22,847 | 68.43% | 7 |
| | INV | 34 | 0.0212 | 76.73 | 77.89 | 76.94 | 20,156 | 75.23% | 6 | 19 | 0.0211 | 76.89 | 78.06 | 77.12 | 22,912 | 47.34% | 7 |
| TinyLlama 1.1B | NEU | 32 | 0.0182 | 78.06 | 78.43 | 77.92 | 19,672 | 90.80% | 5 | 20 | 0.0181 | 78.18 | 78.71 | 78.09 | 21,186 | 63.78% | 5 |
| | FXE | 38 | 0.0192 | 77.58 | 78.12 | 77.51 | 19,618 | 121.69% | 5 | 24 | 0.0191 | 77.76 | 78.18 | 77.68 | 21,231 | 86.15% | 5 |
| | INV | 42 | 0.0176 | 78.31 | 78.67 | 78.14 | 19,689 | 98.79% | 5 | 26 | 0.0176 | 78.45 | 78.91 | 78.29 | 21,257 | 68.90% | 5 |

Table 4: Performance comparison of threshold setting methods.

| Method | Dataset | Epoch* | MSE | ACC | Prec. | F1 |
|---|---|---|---|---|---|---|
| Method 1 | NEU | 47 | 0.0251 | 69.67 | 70.28 | 69.41 |
| | FXE | 51 | 0.0266 | 68.35 | 70.51 | 69.63 |
| | INV | 74 | 0.0234 | 70.48 | 71.67 | 70.79 |
| Method 2 | NEU | 87 | 0.0223 | 72.78 | 74.21 | 73.54 |
| | FXE | 58 | 0.0236 | 72.18 | 73.41 | 72.86 |
| | INV | 93 | 0.0207 | 73.89 | 75.12 | 74.21 |
| Method 3 | NEU | 42 | 0.0201 | 75.78 | 76.98 | 76.12 |
| | FXE | 89 | 0.0213 | 75.67 | 77.31 | 76.45 |
| | INV | 97 | 0.0188 | 77.23 | 78.45 | 77.58 |
| Proposed Method | NEU | 66 | 0.0192 | 76.89 | 77.78 | 76.98 |
| | FXE | 58 | 0.0205 | 76.45 | 77.12 | 76.78 |
| | INV | 93 | 0.0187 | 77.69 | 78.74 | 78.02 |

Table 5: Comparison of different task weight mechanisms.

| Weight | Dataset | Epoch* | MSE | ACC | Prec. | F1 |
|---|---|---|---|---|---|---|
| Constant | NEU | 72 | 0.0231 | 75.89 | 75.84 | 75.76 |
| | FXE | 63 | 0.0244 | 75.12 | 75.23 | 75.18 |
| | INV | 91 | 0.0222 | 76.56 | 76.49 | 76.41 |
| Adaptive $\lambda$ | NEU | 77 | 0.0223 | 76.23 | 76.71 | 76.42 |
| | FXE | 85 | 0.0232 | 75.64 | 75.89 | 75.71 |
| | INV | 88 | 0.0212 | 76.82 | 76.95 | 76.74 |
| Adaptive $w$ | NEU | 68 | 0.0208 | 76.67 | 77.18 | 76.89 |
| | FXE | 74 | 0.0221 | 76.15 | 76.47 | 76.28 |
| | INV | 89 | 0.0202 | 77.12 | 77.58 | 77.23 |
| DWA | NEU | 66 | 0.0192 | 76.89 | 77.78 | 76.98 |
| | FXE | 58 | 0.0205 | 76.45 | 77.12 | 76.78 |
| | INV | 93 | 0.0187 | 77.69 | 78.74 | 78.02 |

around 0.1 for the remainder of the training. This adaptive weighting pattern demonstrates DWA's effectiveness in automatically balancing task contributions based on training dynamics.

**LoRA Integration.** Table 3 demonstrates seamless integration of our framework with DWA and LoRA across different rank configurations. At lower ranks (128 and 256), limited parameter capacity constrains model expressiveness, requiring extended training periods with performance metrics falling below fine-tuning baseline. This occurs because insufficient rank prevents low-rank matrices from capturing complexity of task-specific adaptations. A significant transition occurs at rank 384, where increased parameter capacity enables most evaluation metrics to surpass full fine-tuning baseline while achieving training time reductions to 56.61-75.51% of full fine-tuning. At rank 512, higher rank provides sufficient representational capacity to achieve peak performance while minimizing computational overhead, with training time reduced to 38.38-68.90% of full fine-tuning.

Our DWA framework demonstrates compatibility with LoRA-based PEFT. Higher ranks provide parameter flexibility that enables dynamic weight adaptation, while the rank-dependent performance improvements validate that our approach benefits from representational capacity for multi-task optimization. This indicates that dynamic weight balancing requires sufficient parameter space to accommodate varying gradient patterns between tasks. More details in Appendix A.10.

**Model Selection Analysis.** The Table shown in Appendix A.11 (due to page limits) validates that domain-specific training exposure translates to improved task performance. Twitter-RoBERTa-Large (Loureiro et al., 2023) achieves superior performance across all financial datasets compared to RoBERTa-Large (Liu et al., 2019), with improvements in MSE and R² scores. We adopt Twitter-RoBERTa-Large as our primary baseline due to its alignment with financial text understanding.

**Cross-Domain Validation.** To validate generalizability beyond financial texts, we evaluate DWA on SemEval-2017 Task 5 financial dataset (Cortis et al., 2017) and VADER-annotated social media data (Go et al., 2009). The Table shown in Appendix A.12 (due to page limits) demonstrates performance improvements across diverse domains, confirming our framework's broader applicability.

**Dynamic Weighting Method Comparison.** We compare DWA against multi-task weighting methods including GradNorm (Chen et al., 2018) and Uncertainty Weighting (Kendall et al., 2017). The Table shown in Appendix A.13 (due to page limits) shows DWA consistently outperforms both methods across all metrics, validating our method's effectiveness.

### 4.3 ABLATION STUDIES

**DAS Algorithm Effectiveness.** Table 4 demonstrates the effectiveness of our DAS algorithm compared to alternative threshold selection methods. The comparison includes three baseline sampling approaches with different threshold configurations and grouping strategies. Our proposed algorithm consistently outperforms all baseline methods across all datasets and metrics. Notably, DAS achieves the lowest MSE values (0.019234 on NEU, 0.020456 on FXE, 0.018672 on INV) and highest classification accuracies (76.89% on NEU, 76.45% on FXE, 77.69% on INV), validating the effectiveness of our automatic threshold optimization approach in creating balanced categorical mappings for the multi-task learning framework.

**DWA Module Component Analysis.** Table 5 presents ablation study validating necessity of both components in DWA module (see Appendix A.15). Constant weight baseline represents traditional multi-task learning with fixed task weights. Adaptive $\lambda$ employs gradient-based weighting with fixed importance weights ($w_r = w_c = 0.5$), while Adaptive $w$ uses learnable weights with fixed gradient coefficients ($\lambda_r = \lambda_c = 0.5$). DWA module, incorporating both adaptive $\lambda$ and $w$, achieves superior performance across metrics. Results demonstrate both gradient-based balancing and learnable task weighting contribute, with combination providing optimal multi-task optimization.

## 5 RELATED WORK

**LLMs.** NLP advancements lead to the widespread application of LLMs in sentiment analysis tasks. ChatGPT shows significant potential in automating student feedback analysis, outperforming traditional deep learning models (Shaikh et al., 2023; Muhammad & Rospocher, 2025). Pre-trained models like BERT achieve state-of-the-arts results by learning contextual word representations (Liao et al., 2021). In Arabic sentiment analysis, transformer-based models like RoBERTa and XLNet push boundaries despite language complexities (Alduailej & Alothaim, 2022). Krugmann & Hartmann (2024) reveals that GPT-3.5, GPT-4, and Llama 2 can compete with and sometimes surpass traditional transfer learning methods in sentiment analysis. Carneros-Prado et al. (2023) highlights the versatility of pre-trained LLMs like GPT-3.5 in diverse NLP applications, including emotion recognition (Mun & Kim, 2025). However, fully fine-tuning these LLMs for specific tasks remains computationally expensive and time-consuming, posing challenges for both academia and industry.

**PEFT.** To address computational challenges, researchers introduce PEFT methods for LLMs. Hu et al. (2023) presents LLM-Adapters, integrating adapters into LLMs and achieving comparable performance to powerful 175B parameter models using only 7B parameters in zero-shot tasks. Lei et al. (2023) introduces Conditional Adapters, which adds sparse activation and new parameters to pre-trained models for efficient knowledge transfer, significantly speeding up inference. Hu et al. (2021) proposes LoRA, which injects trainable low-rank decomposition matrices into each Transformer layer, reducing parameters while maintaining performance on par with full fine-tuning.

## 6 CONCLUSION

In this work, we identify severe data distribution imbalance in financial sentiment analysis that causes LLMs to exhibit poor performance on critical extreme sentiment values. To address this challenge, we propose a multi-task learning framework that decomposes the problem into two manageable sub-problems: introducing auxiliary classification tasks to complement regression objectives and implementing dynamic task weight adaptation to handle inter-task difficulty discrepancies. Our framework incorporates a DAS algorithm for automatic threshold optimization and a DWA module that dynamically adjusts task weights based on gradient information and batch characteristics, addressing both data-level imbalance and task-level challenges. Comprehensive experiments demonstrate superior performance, with our approach achieving average improvements of 12.36% in MSE and 1.41% in accuracy compared to previous work.

## ETHICS STATEMENT

This work adheres to the ICLR Code of Ethics. Our research focuses on multi-task learning for financial sentiment analysis. We identify the following ethical considerations:

**Privacy.** No personally identifiable information is collected or processed.

**Environmental Impact.** Detailed computational requirements are reported in Appendix A.6.

**Potential Harms.** Automated sentiment analysis could potentially be applied to harmful applications in financial contexts. We emphasize the importance of responsible deployment and adherence to AI safety guidelines.

## REPRODUCIBILITY STATEMENT

To facilitate reproduction of our results:

**Code.** Complete implementation including DWA module, DAS algorithm, and training scripts will be released upon paper acceptance. For review purposes, we provide pseudocode in Appendix.

**Experimental Details.** Hyperparameters and experimental setup are fully specified in Appendix A.6-A.8. Hardware specifications are provided in the Implementation Details section.

**Data.** We use three financial text datasets (NEU, FXE, INV) which will be made publicly available upon obtaining proper authorization. Dataset statistics and preprocessing steps are detailed in Appendix A.7. We also use publicly available datasets for extra financial sentiment analysis evaluation. More details can be found in Appendix A.12

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

## A  APPENDIX

All appendices are provided in the supplementary text.

