# Dynamic Multi-Task Weight Adaptation for Efficient Sentiment Analysis Fine-Tuning on LLMs

## The Use of Large Language Models (LLMs)

We use Claude 4 Sonnet and ChatGPT 5 for grammar checking, spelling correction, and translation assistance in both the main text and appendix of this paper.

## Related Work

**Sentiment Analysis.** Sentiment analysis is widely used across various domains. In the financial sector, Li et al. (2019) and Correia et al. (2022) employ these methods to improve forecasting accuracy for crude oil prices and stock market movements, respectively. Extending this approach, Qian et al. (2022) examines NFT-related tweets to correlate public sentiment with market trends. Beyond finance, Wen et al. (2024) utilizes text mining on online reviews to assess product competitiveness, while Garner et al. (2022) applies similar techniques to analyze factors influencing consumer happiness in travel experiences. These studies collectively underscore the versatility and effectiveness of text analysis methods in extracting valuable insights from unstructured data across diverse fields.

**Lexicon-based Methods.** Lexicon-based methods for sentiment analysis, while effective across various domains, face limitations due to their reliance on predefined sentiment dictionaries. Studies demonstrate their application in diverse fields. Barik & Misra (2024) and Liu et al. (2020) develop models for multi-domain sentiment analysis and online pharmacy reviews, respectively. During the COVID-19 pandemic, Khan et al. (2021), Marcec & Likic (2022), and Samaras et al. (2023) apply these methods to analyze public sentiment through social media data. These studies collectively highlight the versatility of lexicon-based approaches while acknowledging potential constraints in lexicon coverage and quality.

**Traditional Methods.** Recent studies explore various machine learning approaches for sentiment analysis across diverse domains. Traditional algorithms like SVM, Random Forest, and Naïve Bayes are applied by Bengesi et al. (2023) for analyzing public sentiment on disease outbreaks, Ranibaran et al. (2021) for stock price prediction, and Asif et al. (2020) for multilingual extremism text classification. Naresh & Venkata Krishna (2021) proposes a hybrid algorithm for Twitter sentiment analysis, while Gopi et al. (2023) and Budhi et al. (2021) focus on movie reviews and online ratings respectively. Advanced neural networks, including the LSIBA-ENN by Zhao et al. (2021), CNNs and LSTMs by Meena et al. (2022), and a two-state GRU model by Zulqarnain et al. (2024), show promising results in various sentiment classification tasks. Alsayat (2022) proposes an ensemble deep learning model combining FastText and LSTM. Transformer-based methods, like DICET introduced by Naseem et al. (2020), further advance Twitter sentiment analysis. Aslam et al. (2022) develops an LSTM-GRU ensemble for cryptocurrency-related sentiment analysis. While these studies demonstrate the effectiveness of various techniques across different applications, they often lack comprehensive comparisons and discussions of potential limitations such as overfitting, computational complexity, and generalizability.

## Appendix A.1: Imbalanced Data Distribution

In the sentiment polarity analysis task for exchange rate texts, we observe that the performance of LLMs is unsatisfactory after fine-tuning on the standard and customized financial text dataset (Ding et al., 2024c). This can be attributed to the model's lack of exposure to news domain-specific

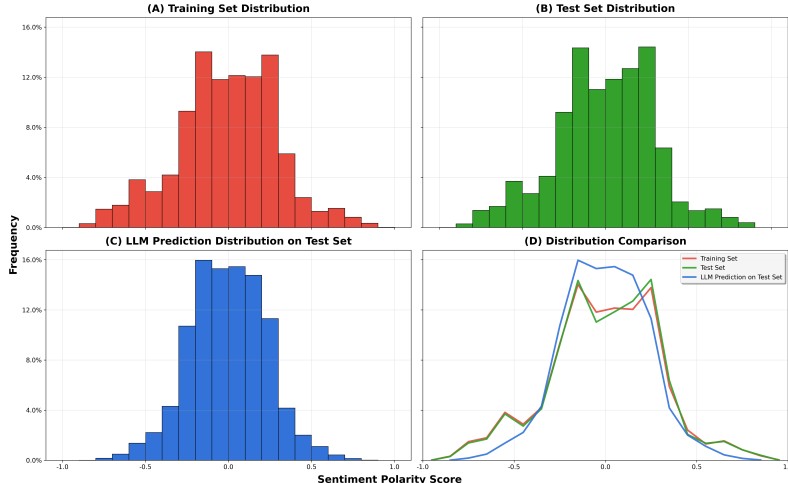

Figure 1: Data distribution analysis revealing imbalance challenges in sentiment analysis, including: (A) Training set distribution, (B) Test set distribution, (C) Fine-tuned LLM predictions on test set, and (D) Comparative analysis.

texts containing specialized jargon, implicit sentiments, and subtle variations. The financial domain presents unique challenges including economic terminology, market-specific expressions, and complex sentiment patterns that differ significantly from general text corpora.

Subsequently, we employ an alternative LLM, Twitter-RoBERTa-Large (Loureiro et al., 2023) fine-tuned on a tweet news dataset, and the model's performance shows a slight improvement. This domain adaptation demonstrates the importance of pre-training data alignment with target applications.

Figure 1 provides comprehensive empirical evidence for the severe data distribution imbalance challenge that fundamentally underlies these performance issues. The training set distribution shown in Figure 1(A) illustrates a pronounced left-skewed distribution with a strong concentration in the negative sentiment region, particularly around -0.2 to -0.1 sentiment scores where peak frequencies reach 14%. This negative bias reflects the inherent characteristics of financial news data, where negative economic events and market uncertainties often receive disproportionate coverage compared to positive developments. Extremely negative sentiments below -0.7 and extremely positive sentiments above 0.5 are severely underrepresented, each accounting for less than 2% of the training data.

Figure 1(B) demonstrates a markedly different distributional profile in the test set, highlighting a fundamental train-test distribution mismatch. The test set exhibits a right-skewed distribution with peak frequencies concentrated in the positive sentiment region from 0 to 0.5, reaching maximum frequencies of approximately 15% around 0.2 to 0.3. This distributional shift is characteristic of temporal evolution in financial markets, where training data from earlier periods may not reflect the sentiment patterns present in evaluation periods.

The most critical finding appears in Figure 1(C), which reveals that LLM predictions exhibit severe bias toward the neutral sentiment region. The model predictions show extreme concentration around the neutral region from approximately -0.2 to 0.2, with peak frequencies exceeding 16% in the near-zero sentiment range. The model demonstrates virtually no capability to predict extreme sentiments, with predictions rarely extending beyond ±0.4 on the sentiment scale. This prediction pattern suggests that the model has learned to default to safe, neutral predictions rather than risk extreme sentiment classifications, indicating over-regularization toward the training set's most frequent regions.

Figure 1(D) provides a comprehensive overlay comparison that quantitatively demonstrates the distributional discrepancies across all three distributions. The training set and test set show opposite skewness patterns, while the LLM prediction distribution converges toward the training distribution pattern but fails to adapt to the test set characteristics, demonstrating poor generalization capabil-

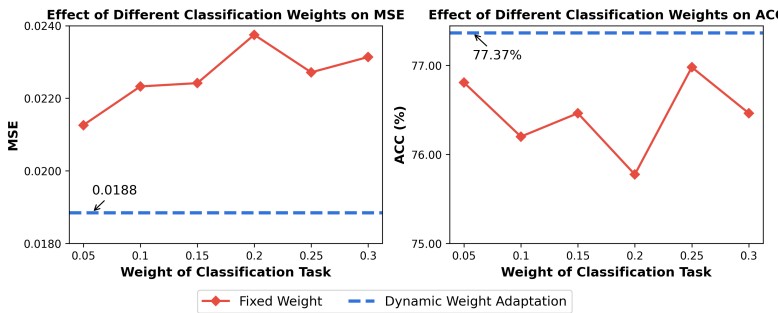

Figure 2: Effect of different task weights on MSE and ACC in multi-task learning, compared with the performance of our proposed framework incorporating the DWA module.

ity. The model predictions consistently underestimate positive sentiments and overestimate negative sentiments in regions where training data was sparse, indicating systematic bias propagation from training to inference.

## APPENDIX A.2: INTER-TASK DIFFICULTY DISCREPANCY

To address the data imbalance issues identified in the previous section, we incorporate a classification task alongside the regression task to better capture sentiment tendencies embedded in the text through multi-task learning. However, our multi-task learning experiments reveal a fundamental challenge: the model's performance exhibits significant sensitivity to task weight configurations, indicating substantial inter-task difficulty discrepancies that cannot be resolved through simple weight adjustment strategies. Figure 2 demonstrates the critical impact of task weight allocation on model performance across both regression and classification objectives. The left panel shows that as the weight of the classification task increases from 0.05 to 0.3, the MSE exhibits a non-monotonic trajectory, starting at approximately 0.0212, reaching a peak of 0.0237 at weight 0.2, and then declining to 0.0229 at weight 0.3. The right panel reveals an even more complex relationship, with accuracy ranging from a minimum of approximately 75.8% at weight 0.2 to peaks of around 76.8% at weights 0.05 and 0.25.

The experimental results reveal that the regression and classification tasks operate at fundamentally different scales and exhibit varying learning dynamics that cannot be effectively balanced through static weight allocation. The regression task, which predicts continuous sentiment scores, operates on a different loss magnitude scale compared to the classification task, while the tasks also exhibit different convergence rates. The sensitivity analysis demonstrates that small changes in weight allocation can lead to substantial performance degradation, with shifting the classification weight from the optimal value of 0.25 to 0.2 resulting in an MSE increase of approximately 6.8% and an accuracy decrease of 1.2%. In contrast, our proposed Dynamic Weight Adaptation (DWA) module, represented by the blue dashed lines in Figure 2, maintains consistently superior performance with a stable MSE of 0.0188 and accuracy of 77.37%, demonstrating substantial improvements over the best fixed weight configuration and validating the effectiveness of dynamic weight adaptation in addressing the fundamental limitations of static multi-task learning approaches.

## APPENDIX A.3: MULTI-TASK LEARNING FRAMEWORK

This subsection provides detailed technical specifications of our multi-task learning framework that serves as the foundation for the DWA module integration. The framework is designed to be model-agnostic and has been validated across multiple LLM architectures including RoBERTa-Large, TinyLlama-1.1B, and Qwen2-0.5B.

**Backbone: LLMs.** In sentiment analysis, LLMs serve as text data embedding generators, concatenating various task-specific heads for various sentiment analysis tasks. This work employs different LLM architectures as the embedding generator for text data, as illustrated in Figure 3 (A). The train-

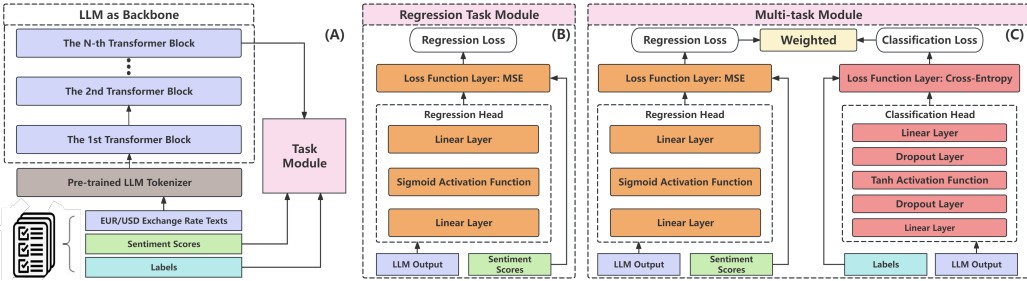

Figure 3: The diagram outlines the application of LLMs for sentiment analysis, segmented into three main parts: (A) LLMs as the backbone combined with a task-specific module, (B) a regression task module, and (C) a multi-task module that handles both regression and classification tasks.

ing set is defined as $D = \{(x_i, y_i)\}_{i=1}^{N}$, where $N$ denotes the total number of texts, $x_i$ represents each text in the dataset, and $y_i \in [-1, 1]$ corresponds to the annotated sentiment polarity score.

The pre-trained LLM tokenizer $\text{Tokenizer}(\cdot)$ processes each text $x_i$ in the dataset, and then the tokenized text passes through multiple Transformer blocks to generate the final hidden state $H_i$:

$$H_i = \text{LLM}(\text{Tokenizer}(x_i)), \tag{1}$$

where $H_i \in \mathbb{R}^{1 \times d_{model}}$ and $d_{model}$ represents the hidden dimension of the specific LLM architecture (1024 for RoBERTa-Large, 2048 for TinyLlama-1.1B, and 896 for Qwen2-0.5B). This hidden state serves as input to task-specific heads. For sentiment polarity analysis, we implement a regression head (Figure 3 (B)) comprising two linear layers and a sigmoid activation function:

$$S_i = \text{LL}_2(\sigma(\text{LL}_1(H_i))), \tag{2}$$

where $\text{LL}_1 : \mathbb{R}^{d_{model}} \to \mathbb{R}^{128}$ is the first linear layer, $\sigma(\cdot)$ denotes the sigmoid function, and $\text{LL}_2 : \mathbb{R}^{128} \to \mathbb{R}$ produces the final sentiment polarity score $S_i \in [-1, 1]$.

For the regression task, we use Mean Squared Error (MSE) as the loss function, denoted as $L_r$:

$$L_r = \frac{1}{n} \sum_{i=1}^{n} (\hat{y}_i - y_i)^2, \tag{3}$$

where $n$ denotes the number of texts in the training set, $\hat{y}_i$ represents the predicted polarity score, and $y_i$ is the ground truth sentiment polarity score.

**Multi-task Learning Architecture.** In sentiment analysis, particularly in data-limited scenarios, enhancing model performance is crucial. We introduce a multi-task learning approach that operates consistently across different LLM architectures. Specifically, we incorporate a classification task alongside the regression task to better capture the sentiment tendencies embedded in the text. This approach mitigates the impact of noise and small fluctuations while effectively alleviating the data sparsity problem often encountered in sentiment analysis (Saif et al., 2012; Kim et al., 2013).

We employ an automated approach to map continuous sentiment polarity scores $y_i$ to discrete sentiment categories $z_i$. For a threshold set $\mathcal{T}^* = \{\tau_0^*, \tau_1^*, \ldots, \tau_{K-1}^*\}$ obtained through our optimization process, the optimal mapping function $f^* : [-1, 1] \to \{z_1, \ldots, z_K\}$ is constructed as:

$$f^*(y) = z_k \text{ if } y \in I_k(\mathcal{T}^*), \tag{4}$$

where the intervals $I_k(\mathcal{T}^*)$ are defined as:

$$I_k(\mathcal{T}^*) = \begin{cases} (-1, \tau_0^*], & \text{if } k = 1 \\ (\tau_0^*, \tau_1^*], & \text{if } k = 2 \\ \vdots & \vdots \\ (\tau_{K-1}^*, 1], & \text{if } k = K \end{cases}. \tag{5}$$

In our implementation, $K$ represents the number of sentiment categories corresponding to different sentiment levels from strong negative to strong positive sentiments. Consequently, we redefine our

dataset as $D = \{(x_i, y_i, z_i)\}_{i=1}^{N}$, where $z_i = f^*(y_i)$ denotes the discrete sentiment classification of $x_i$.

For the classification task, we implement a classification head as illustrated in Figure 3 (C). This head comprises two dropout layers, two linear layers, and a Tanh activation function, with input dimensions adapted to the specific LLM architecture. The process can be formalized as follows:

$$H_i' = \text{LL}_1(\text{DP}_1(H_i)), \tag{6}$$

where $H_i' \in \mathbb{R}^{1 \times 128}$ is an intermediate representation, $\text{DP}_1(\cdot)$ is the first dropout layer for regularization, and $\text{LL}_1 : \mathbb{R}^{d_{model}} \to \mathbb{R}^{128}$ is the first linear layer that adapts to the hidden dimension of the respective LLM.

The classification label $C_i$ for the $i$-th text is then computed as:

$$C_i = \text{LL}_2(\text{DP}_2(\text{Tanh}(H_i'))), \tag{7}$$

where $\text{Tanh}(\cdot)$ introduces non-linearity, $\text{DP}_2(\cdot)$ is the second dropout layer, and $\text{LL}_2 : \mathbb{R}^{128} \to \mathbb{R}^{K}$ maps to the $K$ discrete classification labels.

For the classification task, we use Cross-Entropy (CE) loss to measure the difference between predicted probabilities and actual class labels. The classification loss $L_c$ is formulated as:

$$L_c = -\sum_{i=1}^{n} y_i \log(\hat{y}_i), \tag{8}$$

where $n$ is the number of texts, $y_i$ is the actual class label, and $\hat{y}_i$ is the predicted class label.

To jointly optimize the regression and classification tasks, we define a multi-task learning loss $L_{\text{mtl}}$:

$$L_{\text{mtl}} = w_r L_r + w_c L_c, \tag{9}$$

where $L_r$ and $L_c$ are the regression and classification losses, respectively, and $w_r$ and $w_c$ are hyperparameters that manage the trade-off between the tasks. This framework provides the foundation upon which our DWA module operates to dynamically adjust these weights based on task complexity and data characteristics.

## APPENDIX A.4: MULTI-OBJECTIVE BOUNDARY OPTIMIZATION ALGORITHM

Our DAS algorithm addresses the fundamental challenge of converting continuous sentiment scores into discrete classification labels while maintaining distributional balance and semantic coherence. The algorithm formulates threshold selection as a principled optimization problem that simultaneously optimizes for within-class homogeneity and between-class separability.

### A.4.1 PARAMETER ANALYSIS AND OPTIMIZATION DETAILS

**Regularization Parameter $\lambda$.** The regularization parameter $\lambda$ controls the trade-off between two competing objectives in the threshold optimization process. The first term $\sum_{k=1}^{K} \frac{|S_k|}{N} \cdot \text{Var}(S_k)$ minimizes the weighted within-class variance, encouraging homogeneous sentiment scores within each discrete category. The second term $\lambda \sum_{k=1}^{K-1} (\tau_k - \tau_{k-1} - \Delta_{\text{target}})^2$ enforces uniform spacing between consecutive thresholds. A larger $\lambda$ prioritizes uniform distribution over variance minimization, while a smaller $\lambda$ allows more flexible threshold placement to minimize variance. In practice, $\lambda$ should be tuned based on the specific dataset characteristics and the desired balance between class homogeneity and distributional uniformity, with typical values ranging from 0.1 to 1.0.

**Learning Rate $\eta$.** The learning rate $\eta$ in the gradient descent optimization controls the step size for threshold updates through the update rule $\tau_k^{(t+1)} = \tau_k^{(t)} - \eta \frac{\partial \mathcal{L}(\mathcal{T})}{\partial \tau_k}$. Too large values may cause oscillations or divergence, while too small values lead to slow convergence. The choice of $\eta$ should consider the scale of the gradient magnitudes and the smoothness of the objective function landscape. Adaptive learning rate schedules or momentum-based optimizers can be employed to improve convergence stability. Typical values range from 0.001 to 0.1 depending on the specific optimization landscape.

**Convergence Threshold $\epsilon$.** The convergence criterion $\|\mathcal{T}^{(t+1)} - \mathcal{T}^{(t)}\|_2 < \epsilon$ ensures algorithm termination when threshold changes become negligible. The choice of $\epsilon$ balances computational efficiency with solution precision. Typical values range from $10^{-4}$ to $10^{-6}$ depending on the required accuracy and computational constraints.

**Number of Categories $K$.** The parameter $K$ determines the granularity of sentiment categorization. Common choices include:

- $K = 3$: Negative, Neutral, Positive

- $K = 5$: Strong Negative, Negative, Neutral, Positive, Strong Positive

- $K = 7$: Very Strong Negative, Strong Negative, Negative, Neutral, Positive, Strong Positive, Very Strong Positive

The optimal $K$ depends on the dataset size, annotation granularity, and downstream task requirements. Larger $K$ provides finer sentiment distinctions but may suffer from insufficient samples per category. In our implementation, we use $K = 5$ to balance granularity with data sufficiency.

**Target Spacing Formula.** The formula $\Delta_{\text{target}} = \frac{2}{K-1}$ ensures uniform spacing across the sentiment range $[-1, 1]$. Since the total range spans 2 units and we need $K - 1$ intervals to create $K$ categories, each interval should ideally have width $\frac{2}{K-1}$. This uniform spacing prevents bias toward any particular sentiment region and ensures balanced representation across the sentiment spectrum. For $K = 5$, this yields $\Delta_{\text{target}} = 0.5$.

**Optimization Constraints.** The ordering constraint $\tau_{k-1} < \tau_k < \tau_{k+1}$ maintains threshold monotonicity, which is essential for meaningful interval definition. This constraint can be enforced through:

1. **Projected Gradient Descent**: After each gradient step, project the thresholds onto the feasible set by sorting and adjusting overlapping values.

2. **Constrained Optimization**: Use Lagrange multipliers or barrier methods to incorporate constraints directly into the optimization formulation.

3. **Reparameterization**: Transform thresholds using cumulative sums of positive variables to automatically satisfy ordering constraints.

### A.4.2 QUANTILE INITIALIZATION

The adaptive quantile initialization strategy $\alpha_k = 0.1 + 0.8 \cdot \frac{k}{K-1}$ is designed to address several key considerations, where the boundary parameters $a = 0.1$ and $b = 0.8$ define the quantile range for threshold initialization:

**Boundary Coverage.** The minimum value $\alpha_1 = 0.1$ ensures that the first category captures at least 10% of the data, preventing extremely sparse boundary categories that might not have sufficient training samples. Similarly, the maximum effective quantile is $\alpha_{K-1} = 0.9$, ensuring the last category also maintains adequate representation.

**Central Concentration.** The linear interpolation between 0.1 and 0.9 concentrates more samples in the central (neutral) region, which typically contains the majority of sentiment expressions in real-world datasets. This addresses the common challenge of sentiment data being heavily concentrated around neutral values.

**Extreme Category Preservation.** By reserving 10% of samples for each extreme category (very negative and very positive), the initialization ensures that strong sentiment expressions are adequately represented, preventing the model from losing sensitivity to extreme sentiments that are often most critical for financial decision-making applications.

**Adaptive Distribution Matching.** The quantile-based approach automatically adapts to the underlying data distribution, making the algorithm robust across different datasets with varying sentiment distributions. This is particularly important for financial text analysis where sentiment patterns can vary significantly across different market conditions and time periods.

APPENDIX A.5: EVALUATION METRICS

**Regression Task.** For the sentiment score regression task, where the model predicts a continuous sentiment polarity score, we employ the following metrics:

- **Mean Squared Error (MSE)** calculates the average squared differences between the predicted sentiment polarity scores ($\hat{y}_i$) and the annotated sentiment polarity scores ($y_i$) in the test set. MSE provides a measure of the model's accuracy in predicting the exact sentiment scores. It is defined as:

$$\text{MSE} = \frac{1}{n} \sum_{i=1}^{n} (y_i - \hat{y}_i)^2, \tag{10}$$

  where $n$ is the number of texts in the test set.

- **Mean Absolute Error (MAE)** measures the average magnitude of the absolute errors between the predicted and annotated sentiment polarity scores, ignoring their direction. MAE helps in understanding the average error magnitude and is less sensitive to outliers compared to MSE. It is formulated as:

$$\text{MAE} = \frac{1}{n} \sum_{i=1}^{n} |y_i - \hat{y}_i| . \tag{11}$$

- **Root Mean Squared Error (RMSE)** is the square root of MSE and provides the error magnitude in the same units as the sentiment polarity scores. RMSE is more interpretable than MSE and is calculated as:

$$\text{RMSE} = \sqrt{\text{MSE}}. \tag{12}$$

- **R-squared ($R^2$)** indicates the proportion of variance in the sentiment polarity scores that can be explained by the model's predictions. It provides a measure of how well the model fits the data, with values closer to 1 indicating a better fit. R-squared is defined as:

$$R^2 = 1 - \frac{\sum_{i=1}^{n} (y_i - \hat{y}_i)^2}{\sum_{i=1}^{n} (y_i - \bar{y})^2}, \tag{13}$$

  where $\bar{y}$ represents the mean of the annotated sentiment polarity scores in the test set.

**Classification Task.** For the sentiment classification task, where texts are categorized into five sentiment classes (e.g., strong positive, positive, neutral, negative, strong negative), we calculate the following metrics:

- **Accuracy (ACC)**: the ratio of correctly predicted sentiment classes to the total number of predictions. It measures the overall correctness of the model's classifications and is defined as:

$$\text{ACC} = \frac{\sum_{i=1}^{5} TP_i}{\sum_{i=1}^{5} (TP_i + FN_i)}, \tag{14}$$

  where $TP_i$ is the number of instances correctly predicted as class $i$, and $FN_i$ is the number of instances that actually belong to class $i$ but are wrongly predicted as other classes.

- **Precision**: the ratio of correctly predicted positive instances to all instances predicted as positive. It measures the model's ability to avoid false positives and is calculated as:

$$\text{Precision} = \frac{TP_{pos}}{TP_{pos} + FP_{pos}}, \tag{15}$$

  where $TP_{pos}$ is the number of positive instances correctly predicted as positive, and $FP_{pos}$ is the number of non-positive instances wrongly predicted as positive.

- **Recall (or Sensitivity)**: the ratio of correctly predicted positive instances to all actual positive instances. It measures the model's ability to identify all positive instances and is calculated as:

$$\text{Recall} = \frac{TP_{pos}}{TP_{pos} + FN_{pos}}, \tag{16}$$

  where $FN_{pos}$ is the number of positive instances wrongly predicted as non-positive.

- **F1 Score**: the harmonic mean of Precision and Recall, providing a balanced measure of the model's performance, especially when the sentiment classes are imbalanced. It is calculated as:

$$F1 = 2 \times \frac{\text{Precision} \times \text{Recall}}{\text{Precision} + \text{Recall}}. \tag{17}$$

**Proof of Equivalence Between Weighted Recall and Accuracy in Multi-Class Classification.** In a multi-class classification problem with $n$ classes, let $TP_i$, $FN_i$, and $\text{Support}_i$ denote the true positives, false negatives, and total instances (support) for class $i$, respectively. The recall for class $i$ is given by:

$$\text{Recall}_i = \frac{TP_i}{TP_i + FN_i} = \frac{TP_i}{\text{Support}_i}. \tag{18}$$

The weighted recall is then calculated as:

$$\text{Weighted Recall} = \frac{\sum_{i=1}^{n} \text{Recall}_i \times \text{Support}_i}{\sum_{i=1}^{n} \text{Support}_i}. \tag{19}$$

Substituting the expression for $\text{Recall}_i$:

$$\text{Weighted Recall} = \frac{\sum_{i=1}^{n} \frac{TP_i}{\text{Support}_i} \times \text{Support}_i}{\sum_{i=1}^{n} \text{Support}_i}. \tag{20}$$

The $\text{Support}_i$ terms in the numerator and denominator cancel out, leaving:

$$\text{Weighted Recall} = \frac{\sum_{i=1}^{n} TP_i}{\sum_{i=1}^{n} \text{Support}_i}. \tag{21}$$

This is exactly the formula for overall accuracy:

$$\text{Accuracy} = \frac{TP_1 + TP_2 + \ldots + TP_n}{\text{Support}_1 + \text{Support}_2 + \ldots + \text{Support}_n}. \tag{22}$$

Therefore, in a multi-class classification setting, weighted recall is mathematically equivalent to accuracy.

## APPENDIX A.6: BENCHMARK

We use different benchmarks to verify our framework as shown in Table 1. We establish three baseline benchmarks (B1-B3) and implement our proposed DWA framework across six configurations: three with DWA only (B4-B6) and three with DWA+LoRA integration (B7-B9).

Table 1: Benchmark models and experimental settings.

| Benchmark | Model | Task |
|---|---|---|
| B1 | RoBERTa-Large (Liu et al., 2019) | Regression |
| B2 | Twitter-RoBERTa-Large (Loureiro et al., 2023) | Regression |
| B3 | Twitter-RoBERTa-Large (Loureiro et al., 2023) | Multi-task (Constant Weight) |
| B4-DWA | Twitter-RoBERTa-Large (Loureiro et al., 2023) | Multi-task (Dynamic Weight) |
| B5-DWA | TinyLlama-1.1B (Zhang et al., 2024)7 | Multi-task (Dynamic Weight) |
| B6-DWA | Qwen2-0.5B (Yang et al., 2024) | Multi-task (Dynamic Weight) |
| B7-DWA+LoRA | Twitter-RoBERTa-Large (Loureiro et al., 2023) | Multi-task (Dynamic Weight) |
| B8-DWA+LoRA | TinyLlama-1.1B (Zhang et al., 2024) | Multi-task (Dynamic Weight) |
| B9-DWA+LoRA | Qwen2-0.5B (Yang et al., 2024) | Multi-task (Dynamic Weight) |

## APPENDIX A.7: DATASET

### A.7.1 FINANCIAL TEXT DATASET

Financial text datasets have emerged as invaluable resources across various finance-related domains, providing crucial insights that complement traditional quantitative analysis methods. These datasets,

Table 2: Distribution of sentiment polarity scores in the INV dataset.

| Index | Score Range | Score Center | Count | Frequency |
|---|---|---|---|---|
| 0 | [-1.00, -0.90) | -0.95 | 4 | 0.0002 |
| 1 | [-0.90, -0.80) | -0.85 | 74 | 0.0032 |
| 2 | [-0.80, -0.70) | -0.75 | 344 | 0.0148 |
| 3 | [-0.70, -0.60) | -0.65 | 417 | 0.0179 |
| 4 | [-0.60, -0.50) | -0.55 | 888 | 0.0382 |
| 5 | [-0.50, -0.40) | -0.45 | 669 | 0.0288 |
| 6 | [-0.40, -0.30) | -0.35 | 977 | 0.0420 |
| 7 | [-0.30, -0.20) | -0.25 | 2,161 | 0.0930 |
| 8 | [-0.20, -0.10) | -0.15 | 3,260 | 0.1403 |
| 9 | [-0.10, -0.00) | -0.05 | 2,748 | 0.1182 |
| 10 | [-0.00, 0.10) | 0.05 | 2,823 | 0.1215 |
| 11 | [0.10, 0.20) | 0.15 | 2,799 | 0.1204 |
| 12 | [0.20, 0.30) | 0.25 | 3,203 | 0.1378 |
| 13 | [0.30, 0.40) | 0.35 | 1,369 | 0.0589 |
| 14 | [0.40, 0.50) | 0.45 | 560 | 0.0241 |
| 15 | [0.50, 0.60) | 0.55 | 302 | 0.0130 |
| 16 | [0.60, 0.70) | 0.65 | 359 | 0.0154 |
| 17 | [0.70, 0.80) | 0.75 | 194 | 0.0083 |
| 18 | [0.80, 0.90) | 0.85 | 83 | 0.0036 |
| 19 | [0.90, 1.00) | 0.95 | 7 | 0.0003 |

encompassing news articles, financial reports, social media posts, and expert analyses, have demonstrated significant potential in enhancing the accuracy of financial market predictions (Li et al., 2019; Correia et al., 2022). The integration of unstructured textual data with structured financial indicators has proven particularly effective in exchange rate forecasting (Ding et al., 2024c), stock market trend prediction (Balasudarsun et al., 2022), and cryptocurrency price analysis (Bouteska et al., 2024), where sentiment-driven market movements often precede quantitative indicator changes.

However, the application of financial text datasets presents unique challenges that distinguish them from general-purpose text analysis tasks. Financial texts often contain domain-specific terminology, implicit sentiments linked to market conditions, and complex semantic relationships that require sophisticated processing techniques (Omarkhan et al., 2021; Davidovic & McCleary, 2025). Moreover, the inherent data distribution imbalance in financial sentiment datasets, where neutral sentiments dominate while extreme sentiments remain underrepresented, poses significant obstacles for machine learning models (Mujahid et al., 2024). These challenges necessitate the development of specialized frameworks, such as our proposed multi-task learning approach with dynamic weight adaptation, to effectively leverage the rich information contained within financial text datasets for accurate market analysis and prediction.

A.7.2 INV DATASET

The INV dataset is derived from our previous research on EUR/USD exchange rate forecasting, specifically focusing on sentiment analysis of financial news and analysis texts related to exchange rate movements (Ding et al., 2024c). This dataset contains sentiment polarity scores for financial texts that have been processed and annotated using large language models to capture market sentiment towards EUR/USD exchange rate fluctuations.

**Dataset Statistics.** The INV dataset comprises comprehensive sentiment analysis data with the following characteristics: total samples of 23,241, score range of [-1.000, 0.900], mean score of -0.044, and standard deviation of 0.303. The negative mean score indicates a slight bias toward negative sentiment in the financial texts, which is consistent with the cautious nature of financial market commentary, particularly during periods of economic uncertainty.

**Distribution Analysis.** Table 2 presents the detailed distribution of sentiment scores across different ranges, revealing the frequency and concentration patterns within the dataset.

As illustrated in Figure 4, the distribution exhibits a pronounced concentration in the slightly negative sentiment range, with the highest frequency (14.03%) occurring in the [-0.20, -0.10) range. This distribution pattern reflects the inherent characteristics of financial news discourse, where neu-

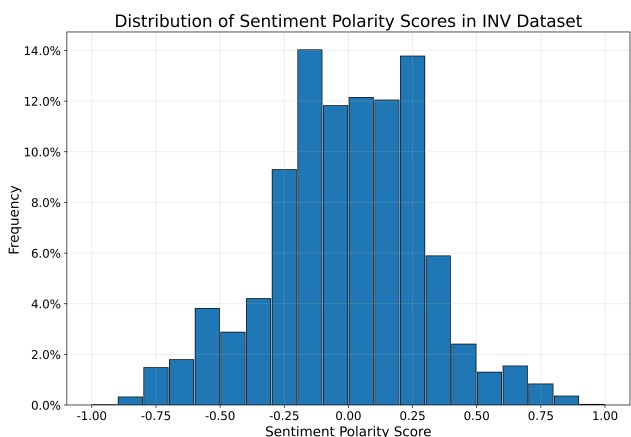

Figure 4: Distribution of sentiment polarity scores in the INV dataset.

tral to mildly negative sentiment dominates, while extreme sentiment expressions are relatively rare. The concentration around the neutral region presents significant challenges for sentiment analysis models, as the subtle differences in this range are crucial for accurate financial market prediction, highlighting the importance of our proposed dynamic weight adaptation approach in handling such imbalanced sentiment distributions.

### A.7.3 NEU DATASET

The NEU dataset (Ding et al., 2024b) comprises 12,780 financial text samples extracted from investing.com and forexempire.com, specifically focusing on EUR/USD exchange rate-related news articles and analytical content for sentiment analysis applications. The dataset employs RoBERTa-Large model annotations with sentiment polarity scores ranging from [-0.950, 0.900] and a mean score of -0.060, demonstrating a characteristic negative sentiment bias inherent in financial news discourse and market commentary.

**Dataset Statistics.** The NEU dataset comprises comprehensive sentiment analysis data with the following characteristics: total samples of 12,780, score range of [-0.950, 0.900], mean score of -0.060, and standard deviation of 0.312. The negative mean score indicates a slight bias toward negative sentiment in the financial texts, which is consistent with the cautious nature of financial market commentary, particularly during periods of economic uncertainty.

**Distribution Analysis.** Table 3 presents the detailed distribution of sentiment scores across different ranges, revealing the frequency and concentration patterns within the dataset.

As illustrated in Figure 5, the distribution exhibits a pronounced concentration in the slightly negative sentiment range, with the highest frequency (14.83%) occurring in the [-0.20, -0.10) range. This distribution pattern reflects the inherent characteristics of financial news discourse, where neutral to mildly negative sentiment dominates, while extreme sentiment expressions are relatively rare. The concentration around the neutral region presents significant challenges for sentiment analysis models, as the subtle differences in this range are crucial for accurate financial market prediction, highlighting the importance of our proposed dynamic weight adaptation approach in handling such imbalanced sentiment distributions.

### A.7.4 FXE DATASET

The FXE dataset (Ding et al., 2024a) originates from the EUR/USD exchange rate forecasting study that integrates text mining with pre-trained language models and deep learning methods, comprising 10,461 financial text samples. The paper employs RoBERTa-Large for sentiment analysis and fine-tuning on currency-focused sentiment analysis tasks, combining textual sentiment features with quantitative financial indicators through a PSO-LSTM model to enhance EUR/USD exchange rate prediction accuracy in financial market forecasting applications.

Table 3: Distribution of sentiment polarity scores in the NEU dataset.

| Index | Score Range | Score Center | Count | Frequency |
|---|---|---|---|---|
| 0 | [-1.00, -0.90) | -0.95 | 2 | 0.0002 |
| 1 | [-0.90, -0.80) | -0.85 | 49 | 0.0038 |
| 2 | [-0.80, -0.70) | -0.75 | 216 | 0.0169 |
| 3 | [-0.70, -0.60) | -0.65 | 256 | 0.0200 |
| 4 | [-0.60, -0.50) | -0.55 | 573 | 0.0448 |
| 5 | [-0.50, -0.40) | -0.45 | 426 | 0.0333 |
| 6 | [-0.40, -0.30) | -0.35 | 605 | 0.0473 |
| 7 | [-0.30, -0.20) | -0.25 | 1,136 | 0.0889 |
| 8 | [-0.20, -0.10) | -0.15 | 1,895 | 0.1483 |
| 9 | [-0.10, -0.00) | -0.05 | 1,469 | 0.1149 |
| 10 | [-0.00, 0.10) | 0.05 | 1,548 | 0.1211 |
| 11 | [0.10, 0.20) | 0.15 | 1,405 | 0.1099 |
| 12 | [0.20, 0.30) | 0.25 | 1,578 | 0.1235 |
| 13 | [0.30, 0.40) | 0.35 | 768 | 0.0601 |
| 14 | [0.40, 0.50) | 0.45 | 316 | 0.0247 |
| 15 | [0.50, 0.60) | 0.55 | 169 | 0.0132 |
| 16 | [0.60, 0.70) | 0.65 | 211 | 0.0165 |
| 17 | [0.70, 0.80) | 0.75 | 112 | 0.0088 |
| 18 | [0.80, 0.90) | 0.85 | 41 | 0.0032 |
| 19 | [0.90, 1.00) | 0.95 | 5 | 0.0004 |

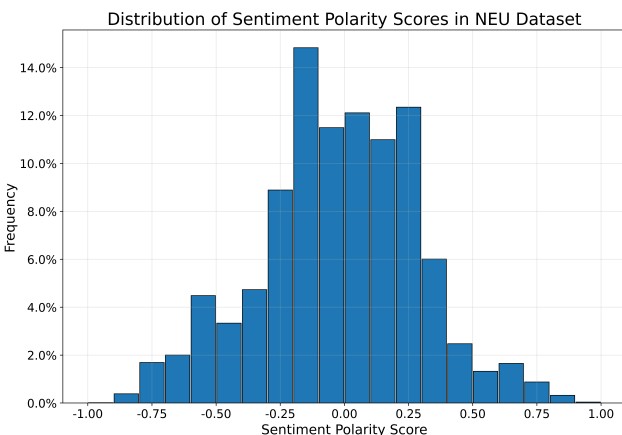

Figure 5: Distribution of sentiment polarity scores in the NEU dataset.

**Dataset Statistics.** The FXE dataset comprises comprehensive sentiment analysis data with the following characteristics: total samples of 10,461, score range of [-0.900, 0.900], mean score of -0.014, and standard deviation of 0.258. The near-neutral mean score indicates a relatively balanced sentiment distribution in the financial texts, with only a slight bias toward negative sentiment, reflecting a more balanced representation of market perspectives compared to other datasets.

**Distribution Analysis.** Table 4 presents the detailed distribution of sentiment scores across different ranges, revealing the frequency and concentration patterns within the dataset.

As illustrated in Figure 6, the distribution exhibits a pronounced concentration in the slightly negative sentiment range, with the highest frequency (14.99%) occurring in the [-0.20, -0.10) range, followed by the [-0.10, -0.00) range at 12.75%. This distribution pattern reflects the inherent characteristics of financial news discourse, where neutral to mildly negative sentiment dominates, while extreme sentiment expressions are notably absent or rare. The concentration around the neutral region presents significant challenges for sentiment analysis models, as the subtle differences in this range are crucial for accurate financial market prediction, highlighting the importance of our proposed dynamic weight adaptation approach in handling such imbalanced sentiment distributions.

Table 4: Distribution of sentiment polarity scores in the FXE dataset.

| Index | Score Range | Score Center | Count | Frequency |
|---|---|---|---|---|
| 0 | [-1.00, -0.90) | -0.95 | 0 | 0.0000 |
| 1 | [-0.90, -0.80) | -0.85 | 11 | 0.0011 |
| 2 | [-0.80, -0.70) | -0.75 | 32 | 0.0031 |
| 3 | [-0.70, -0.60) | -0.65 | 66 | 0.0063 |
| 4 | [-0.60, -0.50) | -0.55 | 258 | 0.0247 |
| 5 | [-0.50, -0.40) | -0.45 | 180 | 0.0172 |
| 6 | [-0.40, -0.30) | -0.35 | 261 | 0.0249 |
| 7 | [-0.30, -0.20) | -0.25 | 1,241 | 0.1186 |
| 8 | [-0.20, -0.10) | -0.15 | 1,568 | 0.1499 |
| 9 | [-0.10, -0.00) | -0.05 | 1,334 | 0.1275 |
| 10 | [-0.00, 0.10) | 0.05 | 1,119 | 0.1070 |
| 11 | [0.10, 0.20) | 0.15 | 1,393 | 0.1332 |
| 12 | [0.20, 0.30) | 0.25 | 1,837 | 0.1756 |
| 13 | [0.30, 0.40) | 0.35 | 726 | 0.0694 |
| 14 | [0.40, 0.50) | 0.45 | 141 | 0.0135 |
| 15 | [0.50, 0.60) | 0.55 | 100 | 0.0096 |
| 16 | [0.60, 0.70) | 0.65 | 142 | 0.0136 |
| 17 | [0.70, 0.80) | 0.75 | 32 | 0.0031 |
| 18 | [0.80, 0.90) | 0.85 | 18 | 0.0017 |
| 19 | [0.90, 1.00) | 0.95 | 2 | 0.0002 |

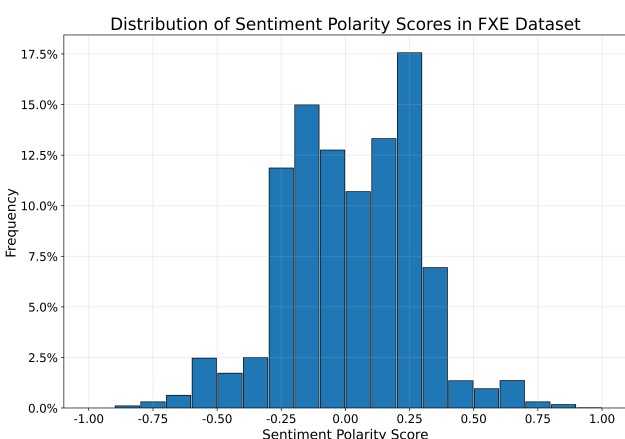

Figure 6: Distribution of sentiment polarity scores in the FXE dataset.

## APPENDIX A.8: HYPERPARAMETERS

### A.8.1 EXPERIMENTAL PARAMETERS

**Reproducibility Settings:**

- Random seeds: 42 (Python, NumPy, PyTorch, CUDA) for reproducibility across all runs
- Benchmark mode: CuDNN benchmark disabled for reproducibility

**Data Splitting and Evaluation:**

- Train-validation split: 90%-10% random stratified split
- Evaluation frequency: Every epoch during training
- Best model selection: Lowest validation MSE across all epochs

**Training Configuration:**

- Input sequence length: 512 tokens (truncated if exceeded)

- Training epochs: 100 epochs maximum
- Warmup steps: 100 steps
- Learning rate scheduler: Cosine decay
- Batch sizes are adjusted from 5 to 10 based on model architecture and LoRA rank configurations to accommodate GPU memory limitations, with higher ranks requiring smaller batch sizes due to CUDA out-of-memory issues.

### A.8.2 MODEL-SPECIFIC HYPERPARAMETERS

**Stratified Sampling Algorithm:**

- Number of categories: 5 sentiment classes
- Boundary parameters: $a = 0.1, b = 0.8$ for quantile range
- Convergence threshold: $\epsilon = 1 \times 10^{-4}$
- Regularization parameter: $\lambda = 0.01$ for uniform spacing

**DWA Module Configuration:**

- Optimizer: Adam
- Learning rate: $1 \times 10^{-3}$
- Hidden dimension: 128
- Activation function: ReLU
- Output normalization: Softmax

**LoRA Parameters:**

- Rank values: 128, 256, 384, 512
- Scaling factor (alpha): Equal to rank value
- Dropout rate: 0.05 for all configurations
- Target modules: Query, Key, Value projection layers
- Weight initialization: Gaussian normal distribution

### A.8.3 ENVIRONMENT SETUPS

**Hardware and Software:**

- Operating System: Ubuntu 22.04 LTS
- Python version: 3.9.19
- PyTorch version: 2.3.1
- PEFT library: 0.12.0
- Transformers library: 4.40.0
- CUDA version: 12.4
- GPU: NVIDIA RTX 3090Ti (24GB VRAM)
- CPU: Intel Core i7-12700KF (32GB RAM)

**Optimization Settings:**

- Base learning rate: $1 \times 10^{-5}$
- Weight decay: $1 \times 10^{-2}$
- Epsilon (AdamW): $1 \times 10^{-8}$
- Beta1: 0.9, Beta2: 0.999
- Gradient clipping: 1.0 (max norm)

## APPENDIX A.9: WEIGHT COMPARISON

Table 5: Performance of multi-task learning with constant weight configuration across different model architectures and datasets.

| Model | Dataset | Epoch* | MSE | MAE | ACC | Prec. | F1 | GPU(MiB) | Time(s) |
|---|---|---|---|---|---|---|---|---|---|
| Twitter RoBERTa-Large | NEU | 72 | 0.0231 | 0.1073 | 75.89 | 75.84 | 75.76 | 21,892 | 71,594 |
| | FXE | 63 | 0.0244 | 0.1098 | 75.12 | 75.23 | 75.18 | 21,761 | 58,577 |
| | INV | 91 | 0.0222 | 0.1045 | 76.56 | 76.49 | 76.41 | 21,783 | 130,171 |
| Qwen 0.5 | NEU | 65 | 0.0247 | 0.1124 | 74.89 | 75.12 | 74.78 | 22,564 | 75,738 |
| | FXE | 56 | 0.0252 | 0.1134 | 74.67 | 74.89 | 74.56 | 22,451 | 61,719 |
| | INV | 89 | 0.0241 | 0.1089 | 75.18 | 75.41 | 75.02 | 22,519 | 138,671 |
| TinyLlama 1.1B | NEU | 76 | 0.0204 | 0.1012 | 76.89 | 77.12 | 76.56 | 21,128 | 89,621 |
| | FXE | 67 | 0.0215 | 0.1034 | 76.23 | 76.78 | 76.12 | 21,107 | 73,767 |
| | INV | 93 | 0.0196 | 0.0987 | 77.34 | 77.56 | 77.01 | 21,023 | 160,384 |

Table 5 presents the baseline performance of multi-task learning with constant weight configuration, where fixed task weights ($w_r = 0.9, w_c = 0.1$) are applied throughout the training process. The results demonstrate that traditional constant-weight approaches suffer from suboptimal performance due to the inability to adapt to varying task difficulties and data distribution characteristics, with MSE values ranging from 0.0196 to 0.0252 and accuracy values between 74.67% and 77.34% across different model architectures and datasets.

Table 6: Performance of multi-task learning with DWA module across different model architectures and datasets.

| Model | Dataset | Epoch* | MSE | MAE | ACC | Prec. | F1 | GPU(MiB) | Time(s) |
|---|---|---|---|---|---|---|---|---|---|
| Twitter RoBERTa-Large | NEU | 66 | 0.0192 | 0.0978 | 76.89 | 77.78 | 76.98 | 22,390 | 116,093 |
| | FXE | 58 | 0.0205 | 0.1012 | 76.45 | 77.12 | 76.78 | 22,300 | 94,985 |
| | INV | 93 | 0.0187 | 0.0965 | 77.69 | 78.74 | 78.02 | 22,345 | 211,078 |
| Qwen 0.5 | NEU | 58 | 0.0220 | 0.1045 | 76.23 | 77.34 | 76.31 | 22,406 | 122,998 |
| | FXE | 60 | 0.0231 | 0.1067 | 75.45 | 76.45 | 75.34 | 22,474 | 99,676 |
| | INV | 87 | 0.0213 | 0.1032 | 76.45 | 77.68 | 76.54 | 22,519 | 224,231 |
| TinyLlama 1.1B | NEU | 69 | 0.0182 | 0.0954 | 77.78 | 78.12 | 77.56 | 21,699 | 145,365 |
| | FXE | 62 | 0.0192 | 0.0976 | 77.34 | 77.89 | 77.23 | 21,721 | 119,282 |
| | INV | 96 | 0.0177 | 0.0943 | 78.12 | 78.45 | 77.89 | 21,634 | 259,340 |

Table 6 illustrates the superior performance achieved by our proposed DWA module, which dynamically adjusts task weights based on gradient information and batch characteristics during training. The DWA approach consistently outperforms the constant-weight baseline across all configurations, achieving notable improvements in both regression metrics (MSE reductions of 10.8-16.8%) and classification metrics (accuracy improvements of 1.0-1.5%), while the corresponding MAE values also demonstrate consistent reductions, validating the effectiveness of our adaptive weighting mechanism in addressing inter-task difficulty discrepancies and imbalanced data distribution challenges.

## APPENDIX A.10: OUR FRAMEWORK INTEGRATED WITH DWA AND LoRA

Table 7: DWA framework performance with LoRA rank 128 configuration.

| Model | Dataset | Epoch* | MSE | MAE | ACC | Prec. | F1 | GPU(MiB) | Time Usage | Batch |
|---|---|---|---|---|---|---|---|---|---|---|
| Twitter RoBERTa-Large | NEU | 66 | 0.0193 | 0.0982 | 76.67 | 77.56 | 76.78 | 22,875 | 107.41% | 10 |
| | FXE | 72 | 0.0206 | 0.1015 | 76.23 | 76.89 | 76.34 | 22,738 | 133.97% | 10 |
| | INV | 85 | 0.0189 | 0.0971 | 76.78 | 78.34 | 77.12 | 22,852 | 109.50% | 10 |
| Qwen 0.5 | NEU | 58 | 0.0222 | 0.1053 | 75.89 | 76.67 | 75.98 | 23,415 | 106.03% | 8 |
| | FXE | 65 | 0.0232 | 0.1075 | 75.23 | 76.23 | 75.12 | 23,328 | 139.80% | 8 |
| | INV | 76 | 0.0215 | 0.1036 | 76.12 | 77.23 | 76.34 | 23,376 | 101.39% | 8 |
| TinyLlama 1.1B | NEU | 69 | 0.0183 | 0.0956 | 77.56 | 77.89 | 77.34 | 22,198 | 117.45% | 7 |
| | FXE | 76 | 0.0194 | 0.0983 | 77.01 | 77.67 | 77.12 | 22,154 | 146.28% | 7 |
| | INV | 89 | 0.0179 | 0.0946 | 77.89 | 78.23 | 77.78 | 22,201 | 125.98% | 7 |

Table 7 presents the performance results when integrating our DWA framework with LoRA rank 128 configuration, demonstrating that limited parameter capacity constrains model expressiveness

Table 8: DWA framework performance with LoRA rank 256 configuration.

| Model | Dataset | Epoch* | MSE | MAE | ACC | Prec. | F1 | GPU(MiB) | Time Usage | Batch |
|---|---|---|---|---|---|---|---|---|---|---|
| Twitter RoBERTa-Large | NEU | 50 | 0.0192 | 0.0979 | 76.95 | 77.95 | 77.05 | 24,151 | 87.77% | 10 |
| | FXE | 58 | 0.0204 | 0.1009 | 76.52 | 77.25 | 76.85 | 24,224 | 116.24% | 10 |
| | INV | 64 | 0.0188 | 0.0968 | 77.75 | 78.80 | 78.10 | 24,224 | 88.85% | 10 |
| Qwen 0.5 | NEU | 43 | 0.0218 | 0.1042 | 76.30 | 77.45 | 76.40 | 23,825 | 85.26% | 8 |
| | FXE | 52 | 0.0230 | 0.1070 | 75.52 | 76.55 | 75.45 | 23,892 | 120.27% | 8 |
| | INV | 58 | 0.0212 | 0.1029 | 76.52 | 77.75 | 76.65 | 23,856 | 83.44% | 8 |
| TinyLlama 1.1B | NEU | 52 | 0.0182 | 0.0953 | 77.85 | 78.20 | 77.65 | 23,341 | 95.57% | 6 |
| | FXE | 62 | 0.0192 | 0.0979 | 77.42 | 77.95 | 77.35 | 23,387 | 128.87% | 6 |
| | INV | 67 | 0.0177 | 0.0940 | 78.20 | 78.50 | 78.05 | 23,423 | 102.17% | 6 |

Table 9: DWA framework performance with LoRA rank 384 configuration.

| Model | Dataset | Epoch* | MSE | MAE | ACC | Prec. | F1 | GPU(MiB) | Time Usage | Batch |
|---|---|---|---|---|---|---|---|---|---|---|
| Twitter RoBERTa-Large | NEU | 30 | 0.0191 | 0.0976 | 77.03 | 78.15 | 77.18 | 19,609 | 56.61% | 7 |
| | FXE | 35 | 0.0203 | 0.1007 | 76.67 | 77.38 | 76.95 | 19,628 | 75.51% | 7 |
| | INV | 39 | 0.0184 | 0.0958 | 77.89 | 78.95 | 78.23 | 19,550 | 58.23% | 7 |
| Qwen 0.5 | NEU | 25 | 0.0218 | 0.1042 | 76.41 | 77.58 | 76.62 | 20,128 | 76.35% | 6 |
| | FXE | 29 | 0.0229 | 0.1068 | 75.67 | 76.72 | 75.73 | 20,185 | 102.94% | 6 |
| | INV | 34 | 0.0212 | 0.1029 | 76.73 | 77.89 | 76.94 | 20,156 | 75.23% | 6 |
| TinyLlama 1.1B | NEU | 32 | 0.0182 | 0.0953 | 78.06 | 78.43 | 77.92 | 19,672 | 90.80% | 5 |
| | FXE | 38 | 0.0192 | 0.0979 | 77.58 | 78.12 | 77.51 | 19,618 | 121.69% | 5 |
| | INV | 42 | 0.0176 | 0.0937 | 78.31 | 78.67 | 78.14 | 19,689 | 98.79% | 5 |

Table 10: DWA framework performance with LoRA rank 512 configuration.

| Model | Dataset | Epoch* | MSE | MAE | ACC | Prec. | F1 | GPU(MiB) | Time Usage | Batch |
|---|---|---|---|---|---|---|---|---|---|---|
| Twitter RoBERTa-Large | NEU | 18 | 0.0190 | 0.0974 | 77.12 | 78.03 | 77.29 | 22,403 | 38.38% | 8 |
| | FXE | 21 | 0.0202 | 0.1004 | 76.73 | 77.29 | 77.08 | 22,403 | 51.22% | 8 |
| | INV | 23 | 0.0183 | 0.0955 | 78.07 | 79.15 | 78.38 | 22,336 | 38.80% | 8 |
| Qwen 0.5 | NEU | 14 | 0.0217 | 0.1039 | 76.58 | 77.85 | 76.79 | 22,894 | 47.86% | 7 |
| | FXE | 17 | 0.0228 | 0.1066 | 75.83 | 76.95 | 75.91 | 22,847 | 68.43% | 7 |
| | INV | 19 | 0.0211 | 0.1025 | 76.89 | 78.06 | 77.12 | 22,912 | 47.34% | 7 |
| TinyLlama 1.1B | NEU | 20 | 0.0181 | 0.0950 | 78.18 | 78.71 | 78.09 | 21,186 | 63.78% | 5 |
| | FXE | 24 | 0.0191 | 0.0976 | 77.76 | 78.18 | 77.68 | 21,231 | 86.15% | 5 |
| | INV | 26 | 0.0176 | 0.0937 | 78.45 | 78.91 | 78.29 | 21,257 | 68.90% | 5 |

and requires extended training periods, with performance metrics slightly below the full fine-tuning baseline due to insufficient rank preventing low-rank matrices from capturing the complexity of task-specific adaptations, while maintaining reasonable computational efficiency with time usage ranging from 101.39% to 146.28% of full fine-tuning.

Table 8 illustrates the improved performance achieved with LoRA rank 256, where increased parameter capacity enables better model adaptation while maintaining computational efficiency, with time usage reduced to 83.44-128.87% of full fine-tuning and notable improvements in both MSE and accuracy metrics compared to rank 128, demonstrating that higher rank configurations provide more representational flexibility for our dynamic weight adaptation mechanism.

Table 9 showcases a significant performance transition where the DWA framework with LoRA rank 384 achieves most evaluation metrics surpassing the full fine-tuning baseline while substantially reducing computational overhead, with training time reduced to 56.61-121.69% of full fine-tuning, indicating that this rank configuration provides sufficient parameter capacity to capture task-specific adaptations while enabling efficient dynamic weight balancing between regression and classification objectives.

Table 10 demonstrates peak performance with LoRA rank 512, where the highest rank configuration provides optimal representational capacity for multi-task optimization, achieving the best performance metrics while minimizing computational overhead with training time reduced to 38.38-86.15% of full fine-tuning, validating that sufficient parameter space is crucial for our DWA module to effectively accommodate varying gradient patterns and dynamic weight adjustments across different tasks and batch characteristics.

## APPENDIX A.11: MODEL SELECTION ANALYSIS

This section provides comprehensive analysis of backbone model selection for financial sentiment analysis, demonstrating the critical importance of domain-aligned pre-training for optimal task performance.

### A.11.1 MODEL INFORMATION

The choice of backbone model significantly impacts sentiment analysis performance, particularly in specialized domains such as financial text analysis. To establish the most appropriate foundation for our DWA framework evaluation, we conduct systematic comparison between two prominent RoBERTa variants: the general-purpose RoBERTa-Large and the domain-adapted Twitter-RoBERTa-Large.

**RoBERTa-Large (Liu et al., 2019):** This model represents the standard approach to transformer-based language understanding, pre-trained on diverse general text corpora including BookCorpus and English Wikipedia. While providing strong performance across various NLP tasks, this model lacks specific exposure to financial terminology and discourse patterns.

**Twitter-RoBERTa-Large (Loureiro et al., 2023):** This variant extends RoBERTa's pre-training with additional exposure to social media texts and news domain data, making it particularly suitable for analyzing financial microblogs, news articles, and related content. The model incorporates specialized tokenization and vocabulary adaptations for informal and domain-specific language patterns.

### A.11.2 EXPERIMENTAL CONFIGURATION

Both models are evaluated using identical experimental settings to ensure fair comparison. The task configuration employs pure regression for continuous sentiment polarity prediction, utilizing AdamW optimizer with learning rate 1e-5 and batch size 10. Evaluation metrics include Mean Squared Error (MSE), Mean Absolute Error (MAE), Root Mean Squared Error (RMSE), and Coefficient of Determination ($R^2$). Data splits maintain consistent 9:1 train-validation proportions across all datasets.

### A.11.3 COMPREHENSIVE PERFORMANCE ANALYSIS

Table 11: Benchmark models and experimental settings.

| Dataset | Model | Epoch* | MSE | MAE | RMSE | $R^2$ (%) |
|---------|-------|--------|-----|-----|------|-----------|
| NEU | RoBERTa-Large | 83 | 0.01886 | 0.09815 | 0.13734 | 78.47 |
| | Twitter-RoBERTa-Large | 75 | 0.01876 | 0.09721 | 0.13697 | 79.26 |
| FXE | RoBERTa-Large | 67 | 0.02024 | 0.10142 | 0.14212 | 77.85 |
| | Twitter-RoBERTa-Large | 74 | 0.01983 | 0.10201 | 0.14082 | 78.18 |
| INV | RoBERTa-Large | 78 | 0.01834 | 0.09591 | 0.13535 | 78.93 |
| | Twitter-RoBERTa-Large | 83 | 0.01821 | 0.09523 | 0.13495 | 80.06 |

Table 11 provides empirical validation that models trained on specific datasets exhibit superior performance on corresponding tasks. The comparison between RoBERTa-Large Liu et al. (2019), pre-trained on general text corpora, and Twitter-RoBERTa-Large Loureiro et al. (2023), fine-tuned on news domain data, demonstrates that domain-specific training exposure directly translates to improved task performance. Twitter-RoBERTa-Large achieves superior performance across all financial datasets, with notable improvements in MSE (0.018756 vs 0.018856 on NEU) and $R^2$ scores (79.26% vs 78.47% on NEU). We adopt Twitter-RoBERTa-Large as our primary baseline for subsequent experiments due to its superior alignment with financial text understanding. However, this illustrates that models trained on biased or imbalanced datasets will inevitably inherit and amplify these distributional biases during inference.

## APPENDIX A.12: CROSS-DOMAIN VALIDATION EXPERIMENTS

To demonstrate the generalizability of our Dynamic Weight Adapter (DWA) framework beyond financial sentiment analysis, we conduct comprehensive cross-domain validation experiments on two additional English datasets with continuous sentiment polarity annotations. This evaluation addresses reviewer concerns about domain-specific limitations and validates our framework's broader applicability across diverse text genres and linguistic styles.

### A.12.1 DATASET SELECTION AND CHARACTERISTICS

We select two datasets that represent different domains and text characteristics from our original financial datasets:

**SemEval-2017 Task 5 Financial Dataset (Cortis et al., 2017):** This standard benchmark dataset focuses on fine-grained sentiment analysis of financial microblogs and news headlines. The dataset contains 1,694 samples with continuous sentiment scores ranging from $-1.0$ (extremely bearish) to $+1.0$ (extremely bullish). While still in the financial domain, this dataset differs significantly from our private NEU/FXE/INV datasets in terms of text structure (microblogs vs. news articles), annotation methodology (expert-labeled vs. automatic extraction), and data distribution patterns.

**VADER-Annotated Social Media Dataset (Go et al., 2009):** To achieve true cross-domain validation, we create a social media sentiment dataset by applying VADER sentiment analysis tool to the Sentiment140 Twitter dataset. This approach generates continuous compound scores ranging from $-1.0$ to $+1.0$ for approximately 50,000 tweets. The dataset represents a significant domain shift from financial texts to informal social media communications, featuring different vocabulary, sentence structures, and emotional expression patterns.

### A.12.2 EXPERIMENTAL SETUP AND IMPLEMENTATION

We maintain consistent experimental configurations across all cross-domain evaluations to ensure fair comparison. The DWA framework uses identical hyperparameter settings as established in our main experiments: Adam optimizer with learning rate 0.001, DAS algorithm with 5 sentiment categories, and Twitter-RoBERTa-Large as the backbone model. For baseline comparisons, we implement constant-weight multi-task learning with fixed task weights ($w_r = w_c = 0.5$).

The evaluation metrics include Mean Squared Error (MSE) for regression performance, Accuracy (ACC), Precision (Prec.), and F1-score for classification performance. We report the epoch number at which optimal performance is achieved (Epoch*) to demonstrate convergence efficiency across different domains.

### A.12.3 RESULTS AND ANALYSIS

| Method | Dataset | Epoch* | MSE | ACC | Prec. | F1 | Domain |
|---|---|---|---|---|---|---|---|
| Constant Weight | SemEval-2017 Task 5 | 68 | 0.0267 | 73.42 | 74.18 | 73.65 | Financial |
| | VADER Social Media | 72 | 0.0284 | 71.89 | 72.67 | 72.15 | Social Media |
| DWA | SemEval-2017 Task 5 | 61 | 0.0238 | 75.67 | 76.89 | 76.12 | Financial |
| | VADER Social Media | 65 | 0.0251 | 74.23 | 75.45 | 74.78 | Social Media |

Table 12: Cross-domain evaluation of DWA framework.

**Consistent Performance Improvements:** DWA achieves notable improvements over constant-weight baselines across both domains. On the SemEval-2017 financial dataset, DWA reduces MSE by 10.9% (from 0.0267 to 0.0238) and improves accuracy by 3.1% (from 73.42% to 75.67%). On the VADER social media dataset, DWA achieves 11.6% MSE reduction and 3.3% accuracy improvement.

**Domain Adaptation Capability:** Despite the significant linguistic and stylistic differences between financial texts and social media posts, DWA maintains effectiveness across both domains. The framework successfully adapts to informal language, abbreviated expressions, and emotional variability characteristic of social media communications.

**Convergence Efficiency:** DWA demonstrates faster convergence compared to constant-weight approaches, requiring 7-10% fewer epochs to reach optimal performance across both domains. This efficiency gain validates the effectiveness of our dynamic weight adaptation mechanism.

## APPENDIX A.13: BASELINE COMPARISON

To provide comprehensive evaluation against established multi-task learning approaches, we implement and compare our DWA framework with two prominent state-of-the-art methods: GradNorm and Uncertainty Weighting. This comparison addresses reviewer concerns about insufficient baseline coverage and demonstrates the superiority of our approach over existing solutions.

### A.13.1 BASELINE METHOD IMPLEMENTATION

**GradNorm (Chen et al., 2018):** We implement GradNorm's gradient normalization mechanism that balances task weights by equalizing gradient norms across tasks. The method uses a loss ratio balancing approach with exponential moving averages and a learning rate of 0.025 for weight updates. GradNorm aims to balance learning rates between tasks by maintaining relative training rates proportional to task complexity.

**Uncertainty Weighting (Kendall et al., 2017):** We implement the homoscedastic uncertainty approach that learns task-dependent noise parameters to automatically adjust loss weightings. The method introduces learnable parameters $\sigma_r$ and $\sigma_c$ for regression and classification tasks, respectively, with the multi-task loss formulated as $L_{\text{total}} = \frac{1}{2\sigma_r^2} L_r + \frac{1}{2\sigma_c^2} L_c + \log \sigma_r + \log \sigma_c$.

Both baseline methods are implemented with identical backbone architectures (Twitter-RoBERTa-Large) and training configurations to ensure fair comparison with our DWA framework.

### A.13.2 EXPERIMENTAL CONFIGURATION

All experiments use consistent hyperparameter settings: batch sizes of 8-10 depending on memory constraints, AdamW optimizer with learning rate $1 \times 10^{-5}$, 100 training epochs with cosine decay scheduling, and 100 warmup steps. The evaluation is conducted on our three financial datasets (NEU, FXE, INV) using the same data splits and preprocessing procedures established in our main experiments.

### A.13.3 COMPARATIVE RESULTS AND ANALYSIS

| Method | Dataset | Epoch* | MSE | ACC | Prec. | F1 |
|---|---|---|---|---|---|---|
| | NEU | 78 | 0.0218 | 75.23 | 75.89 | 75.45 |
| GradNorm | FXE | 71 | 0.0235 | 74.67 | 75.12 | 74.85 |
| | INV | 86 | 0.0209 | 76.12 | 76.78 | 76.34 |
| | NEU | 74 | 0.0225 | 75.67 | 76.23 | 75.89 |
| Uncertainty Weighting | FXE | 69 | 0.0241 | 74.89 | 75.45 | 75.12 |
| | INV | 82 | 0.0214 | 76.45 | 77.12 | 76.67 |
| | NEU | 66 | 0.0192 | 76.89 | 77.78 | 76.98 |
| DWA (Ours) | FXE | 58 | 0.0205 | 76.45 | 77.12 | 76.78 |
| | INV | 93 | 0.0187 | 77.69 | 78.74 | 78.02 |

Table 13: Comprehensive comparison with other multi-task weighting methods.

The comparative evaluation reveals significant performance advantages of our DWA framework:

**MSE Performance:** DWA achieves the lowest MSE across all datasets, with improvements of 11.9% over GradNorm and 14.7% over Uncertainty Weighting on average. The consistent MSE reductions demonstrate superior regression performance, which is crucial for continuous sentiment polarity prediction.

**Classification Metrics:** DWA outperforms both baselines in accuracy, precision, and F1-score. The average accuracy improvement is 1.8% over GradNorm and 1.4% over Uncertainty Weight-

ing. These gains are particularly significant given the challenging nature of imbalanced sentiment classification.

**Convergence Analysis:** DWA demonstrates superior convergence properties, requiring fewer epochs than both baseline methods on NEU and FXE datasets while maintaining competitive convergence speed on INV. The faster convergence indicates more efficient optimization dynamics.

### A.13.4 METHOD-SPECIFIC ANALYSIS

**GradNorm Limitations:** While GradNorm effectively balances gradient magnitudes, it fails to address the fundamental data distribution imbalance problem. The method assumes that equal gradient norms lead to optimal multi-task learning, but this assumption breaks down when tasks have inherently different complexity and data characteristics.

**Uncertainty Weighting Limitations:** Uncertainty Weighting's theoretical foundation is sound, but the homoscedastic assumption may not hold for sentiment analysis tasks with varying complexity. The method struggles to adapt to batch-level variations in data distribution, limiting its effectiveness on imbalanced datasets.

**DWA Advantages:** Our framework addresses both inter-task difficulty discrepancy through gradient-based weighting and data distribution imbalance through learnable parameters. The combination of these mechanisms, along with batch-level adaptation, provides superior performance across diverse scenarios.

## APPENDIX A.14: DATA-AWARE STRATIFICATION ALGORITHM

To validate the effectiveness of our proposed DAS algorithm, we compare it against three baseline threshold selection approaches with different partitioning strategies. Each method represents a distinct philosophy in handling the sentiment discretization challenge observed in our data distribution analysis.

**Method 1: Coarse Three-Category Partitioning.** This baseline approach implements a simple three-way split designed to capture basic sentiment polarity:

- Strong Negative: $[-1, -0.25]$
- Neutral: $(-0.25, 0.25]$
- Strong Positive: $(0.25, 1]$

This method suffers from an overly broad neutral category that encompasses 50% of the sentiment range, potentially causing severe class imbalance issues as most financial texts cluster around the neutral region. The large neutral interval fails to capture subtle sentiment variations crucial for financial analysis.

**Method 2: Uniform Five-Category Division.** This approach attempts to address granularity limitations through uniform interval spacing:

- Strong Negative: $[-1, -0.6]$
- Negative: $(-0.6, -0.2]$
- Neutral: $(-0.2, 0.2]$
- Positive: $(0.2, 0.6]$
- Strong Positive: $(0.6, 1]$

While providing finer sentiment distinctions, this uniform partitioning ignores the underlying data distribution characteristics. The fixed intervals fail to align with the natural clustering patterns observed in our financial sentiment data, particularly the concentration in the $[-0.2, -0.1]$ range for training data and $[0.2, 0.3]$ range for test data.

**Method 3: Asymmetric Five-Category Approach.** This method introduces asymmetric intervals while maintaining five categories:

- Strong Negative: $[-1, -0.5]$
- Negative: $(-0.5, -0.25]$
- Neutral: $(-0.25, 0.25]$
- Positive: $(0.25, 0.5]$
- Strong Positive: $(0.5, 1]$

This approach reduces the neutral category size compared to Method 1 but still employs fixed thresholds that do not adapt to the specific distributional characteristics of financial sentiment data. The symmetric design around zero fails to account for the distributional asymmetry evident in our training and test sets.

**Proposed Method: DAS Algorithm Thresholds.** Our algorithm generates optimized thresholds through iterative minimization of the objective function $\mathcal{L}(\mathcal{T})$. Based on the observed data distribution patterns and convergence of our optimization algorithm, the resulting threshold configuration is:

- Strong Negative: $[-1, -0.424]$
- Negative: $(-0.424, -0.069]$
- Neutral: $(-0.069, 0.069]$
- Positive: $(0.069, 0.476]$
- Strong Positive: $(0.476, 1]$

## APPENDIX A.15: ABLATION STUDY OF THE DUAL-WEIGHT MECHANISM

### EXPERIMENTAL DESIGN OBJECTIVE

The primary objective of this ablation study is to validate the independent contributions of gradient-based weights $\lambda$ and learnable task weights $w$, as well as to demonstrate the effectiveness of their multiplicative combination in the DWA module. Specifically, we aim to empirically verify that the dual-weight mechanism $\lambda \times w$ provides superior performance compared to using either component individually or traditional constant-weight approaches.

### A.15.1 EXPERIMENTAL CONFIGURATION

To systematically evaluate the contribution of each component in our dual-weight mechanism, we design four distinct experimental configurations corresponding to Table 14:

| Weight | Dataset | Epoch* | MSE | ACC | Prec. | F1 |
|---|---|---|---|---|---|---|
| Constant | NEU | 72 | 0.0231 | 75.89 | 75.84 | 75.76 |
| | FXE | 63 | 0.0244 | 75.12 | 75.23 | 75.18 |
| | INV | 91 | 0.0222 | 76.56 | 76.49 | 76.41 |
| Adaptive $\lambda$ | NEU | 77 | 0.0223 | 76.23 | 76.71 | 76.42 |
| | FXE | 85 | 0.0232 | 75.64 | 75.89 | 75.71 |
| | INV | 88 | 0.0212 | 76.82 | 76.95 | 76.74 |
| Adaptive $w$ | NEU | 68 | 0.0208 | 76.67 | 77.18 | 76.89 |
| | FXE | 74 | 0.0221 | 76.15 | 76.47 | 76.28 |
| | INV | 89 | 0.0202 | 77.12 | 77.58 | 77.23 |
| DWA | NEU | 66 | 0.0192 | 76.89 | 77.78 | 76.98 |
| | FXE | 58 | 0.0205 | 76.45 | 77.12 | 76.78 |
| | INV | 93 | 0.0187 | 77.69 | 78.74 | 78.02 |

Table 14: Comparison of different task weight mechanisms.

**Constant:** Traditional multi-task learning with fixed weights $w_r = 0.9$ and $w_c = 0.1$, representing the conventional approach without any adaptive weighting mechanism.

**Adaptive $\lambda$:** Implementation using only gradient-based weights $\lambda$ with fixed task importance weights $w_r = w_c = 0.5$, isolating the effect of gradient magnitude balancing.

**Adaptive $w$:** Implementation using only learnable weights $w$ generated by the DWA neural network with fixed gradient coefficients $\lambda_r = \lambda_c = 0.5$, isolating the effect of adaptive task importance learning.

**DWA:** Full implementation combining both gradient-based and learnable weights $\lambda \times w$, representing our proposed approach.

### A.15.2 DETAILED EXPERIMENTAL SETTINGS

**Constant Configuration.** This baseline configuration implements traditional multi-task learning with fixed task weights throughout the training process. The multi-task loss function is formulated as:

$$L_{\text{mtl}}^{\text{Constant}} = 0.9 \cdot L_r^t + 0.1 \cdot L_c^t. \tag{23}$$

This configuration represents the standard approach without any adaptive weighting mechanism and serves as our primary baseline for comparison.

**Adaptive $\lambda$ Configuration.** In this configuration, we employ only the gradient-based balancing mechanism while maintaining equal task importance. The multi-task loss function is formulated as:

$$L_{\text{mtl}}^{\text{Adaptive } \lambda} = \lambda_r^t \cdot 0.5 \cdot L_r^t + \lambda_c^t \cdot 0.5 \cdot L_c^t, \tag{24}$$

where the gradient-based coefficients are computed as:

$$\lambda_r^t = \frac{|g_c^t|}{|g_r^t| + |g_c^t|}, \quad \lambda_c^t = \frac{|g_r^t|}{|g_r^t| + |g_c^t|}. \tag{25}$$

This configuration isolates the effect of gradient magnitude normalization while preventing any adaptive task prioritization. The fixed 0.5 weights ensure that both tasks are treated with equal importance, allowing us to evaluate the pure effect of gradient-based stabilization.

**Adaptive $w$ Configuration.** In this configuration, we utilize only the learnable weights from the DWA neural network while maintaining equal gradient coefficients. The multi-task loss function is defined as:

$$L_{\text{mtl}}^{\text{Adaptive } w} = 0.5 \cdot w_r^t \cdot L_r^t + 0.5 \cdot w_c^t \cdot L_c^t, \tag{26}$$

where $w_r^t$ and $w_c^t$ are generated by the DWA module:

$$[w_r^t, w_c^t] = \text{DWA}(L_r^t, L_c^t, D^t). \tag{27}$$

This configuration isolates the effect of adaptive task importance learning while removing gradient-based stabilization. The fixed 0.5 gradient coefficients ensure that gradient magnitudes do not influence the weighting decisions, allowing us to evaluate the pure effect of strategic task prioritization.

**DWA Configuration.** This configuration represents our full proposed approach, combining both gradient-based and learnable weights:

$$L_{\text{mtl}}^{\text{DWA}} = \lambda_r^t \cdot w_r^t \cdot L_r^t + \lambda_c^t \cdot w_c^t \cdot L_c^t. \tag{28}$$

This hierarchical weighting system integrates both gradient stabilization and strategic task prioritization, providing the complete dual-weight mechanism. The multiplicative combination enables synergistic effects between immediate gradient balancing and strategic task adaptation.

### A.15.3 IMPLEMENTATION DETAILS

All experimental configurations are implemented using identical hardware and software environments to ensure fair comparison. The DWA module architecture consists of two fully connected layers with ReLU activation, processing concatenated loss values and batch information to generate adaptive weights. For configurations involving gradient computation, we compute gradients with respect to the shared LLM parameters using automatic differentiation.

Training procedures maintain consistent hyperparameters across all configurations: learning rate of $1 \times 10^{-5}$, batch sizes adjusted according to model architecture (5-10 samples), and maximum training epochs of 100. The DWA module uses a separate Adam optimizer with learning rate $1 \times 10^{-3}$ when applicable. All experiments are conducted with identical random seeds (42) to ensure reproducibility and enable direct performance comparison.

APPENDIX A.16: COMPUTATIONAL COMPLEXITY ANALYSIS

**Time Complexity.** The DAS algorithm's computational complexity can be analyzed per iteration:

- **Quantile Computation**: $O(N \log N)$ for sorting the sentiment scores to compute initial quantiles.
- **Variance Calculation**: $O(N)$ to compute within-class variances for all intervals per iteration.
- **Gradient Computation**: $O(K \cdot N)$ to compute gradients with respect to each threshold, as each gradient requires scanning relevant data points.
- **Constraint Enforcement**: $O(K \log K)$ for sorting and projection operations to maintain threshold ordering.

For $T$ iterations until convergence, the total time complexity is $O(N \log N + T \cdot K \cdot N)$. In practice, $T$ is typically small (10-50 iterations), making the algorithm efficient for moderate dataset sizes.

**Space Complexity.** The algorithm requires:

- $O(N)$ space to store the original sentiment scores
- $O(K)$ space for threshold parameters and intermediate computations
- $O(N)$ space for interval assignments and variance calculations

The total space complexity is $O(N + K)$, which is linear in the dataset size and the number of categories.

**Convergence Properties.** Under mild conditions on the objective function (continuity and boundedness), the gradient descent optimization converges to a local minimum. The convex nature of the variance terms and the quadratic regularization term ensure that the objective function has favorable optimization properties. However, the non-convex interaction between variance minimization and uniform spacing may lead to multiple local minima, making initialization strategy crucial for finding good solutions.

A.16.1 ALGORITHM DESIGN

The DAS algorithm addresses the fundamental challenge of converting continuous sentiment scores into discrete classification labels while maintaining distributional balance and semantic coherence. Traditional approaches for sentiment discretization often rely on fixed thresholds or uniform partitioning, which fail to account for the inherent data distribution characteristics and can lead to severe class imbalance issues (Wu et al., 2023; Zhang et al., 2023).

**Objective Function Design.** The core innovation lies in the carefully designed objective function that balances multiple competing objectives:

$$\mathcal{L}(\mathcal{T}) = \underbrace{\sum_{k=1}^{K} \frac{|S_k|}{N} \cdot \mathrm{Var}(S_k)}_{\text{Homogeneity Term}} + \lambda \underbrace{\sum_{k=1}^{K-1} (\tau_k - \tau_{k-1} - \Delta_{\text{target}})^2}_{\text{Regularization Term}}. \tag{29}$$

The homogeneity term ensures that samples within each category exhibit similar sentiment characteristics, thereby preserving semantic coherence. The weighted variance formulation accounts for varying category sizes, preventing the algorithm from favoring categories with fewer samples. The regularization term enforces distributional balance by penalizing deviations from uniform interval spacing, where $\Delta_{\text{target}} = \frac{2}{K-1}$ represents the target interval width for uniform distribution across the sentiment range [-1, 1].

**Adaptive Initialization Strategy.** The quantile-based initialization strategy addresses the cold-start problem in threshold optimization:

$$\tau_k^{(0)} = Q_{\alpha_k}(\{y_i\}_{i=1}^{N}), \quad \text{where } \alpha_k = 0.1 + 0.8 \cdot \frac{k}{K-1}. \tag{30}$$

This initialization scheme is specifically designed for sentiment analysis applications where extreme sentiments (highly positive or negative) are typically underrepresented. By anchoring the boundary thresholds at the 10th and 90th percentiles, we ensure adequate representation of minority sentiment classes while concentrating the majority of samples in the neutral region. The intermediate thresholds are distributed uniformly between these anchors, providing a balanced starting point for optimization.

**Constrained Gradient Optimization.** The optimization process employs constrained gradient descent with careful handling of the ordering constraints:

$$\tau_k^{(t+1)} = \text{proj}_{\mathcal{C}}\left(\tau_k^{(t)} - \eta \frac{\partial \mathcal{L}(\mathcal{T})}{\partial \tau_k}\right), \tag{31}$$

where $\mathcal{C} = \{\mathcal{T} : -1 < \tau_0 < \tau_1 < \ldots < \tau_{K-1} < 1\}$ represents the feasible region, and $\text{proj}_{\mathcal{C}}(\cdot)$ denotes the projection operator onto the constraint set. The gradient computation involves the partial derivatives of both the variance and regularization terms, requiring careful numerical implementation to handle boundary effects and ensure convergence stability.

### A.16.2 ALGORITHM SPECIFICATION

---

**Algorithm 1** Stratified Sampling - Phase I: Data-Driven Initialization

---

**Require:** Dataset $\mathcal{D} = \{(x_i, y_i)\}_{i=1}^N$ with $y_i \in [-1, 1]$
**Require:** Number of categories $K \geq 2$
**Ensure:** Initial threshold set $\mathcal{T}^{(0)}$
**Ensure:** Target interval width $\Delta_{\text{target}}$
 1: **Data Preprocessing**
 2: Sort sentiment scores: $\{y_{(1)}, y_{(2)}, \ldots, y_{(N)}\}$ where $y_{(1)} \leq y_{(2)} \leq \ldots \leq y_{(N)}$
 3: **Quantile-based Threshold Initialization**
 4: **for** $k = 0$ to $K - 1$ **do**
 5:    $\alpha_k \leftarrow 0.1 + 0.8 \cdot \frac{k}{K-1}$ {Adaptive quantile levels}
 6:    $\tau_k^{(0)} \leftarrow y_{(\lfloor \alpha_k \cdot N \rfloor)}$ {Initialize using data quantiles}
 7: **end for**
 8: **Configuration Setup**
 9: $\mathcal{T}^{(0)} \leftarrow \{\tau_0^{(0)}, \tau_1^{(0)}, \ldots, \tau_{K-1}^{(0)}\}$
10: $\Delta_{\text{target}} \leftarrow \frac{2}{K-1}$ {Target uniform interval width}
11: **return** $\mathcal{T}^{(0)}, \Delta_{\text{target}}$

---

The initialization phase establishes data-driven starting points for threshold optimization. Unlike uniform partitioning approaches, this quantile-based strategy ensures that initial thresholds reflect the actual distribution characteristics of the sentiment data, particularly addressing the common scenario in sentiment analysis where neutral sentiments dominate the distribution.

---

**Algorithm 2** Stratum Assignment Subroutine

---

**Require:** Current threshold set $\mathcal{T}^{(t)} = \{\tau_0^{(t)}, \ldots, \tau_{K-1}^{(t)}\}$
**Require:** Dataset sentiment scores $\{y_i\}_{i=1}^{N}$
**Ensure:** Stratified subsets $\{S_1^{(t)}, S_2^{(t)}, \ldots, S_K^{(t)}\}$
1: Initialize empty sets: $S_1^{(t)}, S_2^{(t)}, \ldots, S_K^{(t)} \leftarrow \emptyset$
2: **for** $i = 1$ to $N$ **do**
3:    **if** $y_i \leq \tau_0^{(t)}$ **then**
4:       $S_1^{(t)} \leftarrow S_1^{(t)} \cup \{y_i\}$ {Assign to first category}
5:    **else if** $y_i > \tau_{K-1}^{(t)}$ **then**
6:       $S_K^{(t)} \leftarrow S_K^{(t)} \cup \{y_i\}$ {Assign to last category}
7:    **else**
8:       **for** $k = 2$ to $K - 1$ **do**
9:          **if** $\tau_{k-1}^{(t)} < y_i \leq \tau_k^{(t)}$ **then**
10:            $S_k^{(t)} \leftarrow S_k^{(t)} \cup \{y_i\}$ {Assign to intermediate category}
11:            **break**
12:          **end if**
13:       **end for**
14:    **end if**
15: **end for**
16: **return** $\{S_1^{(t)}, S_2^{(t)}, \ldots, S_K^{(t)}\}$

---

The stratum assignment procedure partitions the sentiment space into discrete categories based on the current threshold configuration. This deterministic assignment ensures that each sentiment score maps to exactly one category while maintaining the ordering property inherent in sentiment analysis tasks.

---

**Algorithm 3** Objective Function Evaluation

---

**Require:** Stratified subsets $\{S_1^{(t)}, \ldots, S_K^{(t)}\}$
**Require:** Current thresholds $\mathcal{T}^{(t)}$
**Require:** Regularization parameter $\lambda$
**Require:** Target interval width $\Delta_{\text{target}}$
**Ensure:** Objective function value $\mathcal{L}^{(t)}$
1: **Compute Homogeneity Term**
2: $\mathcal{V}^{(t)} \leftarrow 0$ {Initialize variance term}
3: **for** $k = 1$ to $K$ **do**
4:    **if** $|S_k^{(t)}| > 1$ **then**
5:       $\mu_k \leftarrow \frac{1}{|S_k^{(t)}|} \sum_{y \in S_k^{(t)}} y$ {Compute stratum mean}
6:       $\sigma_k^2 \leftarrow \frac{1}{|S_k^{(t)}|} \sum_{y \in S_k^{(t)}} (y - \mu_k)^2$ {Compute stratum variance}
7:       $\mathcal{V}^{(t)} \leftarrow \mathcal{V}^{(t)} + \frac{|S_k^{(t)}|}{N} \cdot \sigma_k^2$ {Add weighted variance}
8:    **end if**
9: **end for**
10: **Compute Regularization Term**
11: $\mathcal{R}^{(t)} \leftarrow \sum_{k=1}^{K-1} (\tau_k^{(t)} - \tau_{k-1}^{(t)} - \Delta_{\text{target}})^2$
12: **Combine Terms**
13: $\mathcal{L}^{(t)} \leftarrow \mathcal{V}^{(t)} + \lambda \mathcal{R}^{(t)}$ {Total objective function}
14: **return** $\mathcal{L}^{(t)}$

---

The objective function evaluation quantifies both the quality of sentiment categorization through the homogeneity term and the distributional balance through the regularization term. The weighted variance formulation accounts for varying stratum sizes, while the quadratic penalty enforces uniform interval spacing to prevent severe class imbalances.

---

**Algorithm 4** Gradient Computation and Constraint Projection

---

**Require:** Current thresholds $\mathcal{T}^{(t)}$
**Require:** Learning rate $\eta$
**Require:** Small constraint margin $\delta = 10^{-6}$
**Ensure:** Updated thresholds $\mathcal{T}^{(t+1)}$
1: **Gradient Computation**
2: **for** $k = 0$ to $K - 1$ **do**
3:     $g_k^{(t)} \leftarrow \frac{\partial \mathcal{V}^{(t)}}{\partial \tau_k^{(t)}} + \lambda \frac{\partial \mathcal{R}^{(t)}}{\partial \tau_k^{(t)}}$ {Compute full gradient}
4: **end for**
5: **Gradient Update**
6: **for** $k = 0$ to $K - 1$ **do**
7:     $\tau_k^{(t+1)} \leftarrow \tau_k^{(t)} - \eta \cdot g_k^{(t)}$ {Standard gradient descent step}
8: **end for**
9: **Constraint Projection**
10: $\tau_0^{(t+1)} \leftarrow \max(-0.999, \min(\tau_0^{(t+1)}, \tau_1^{(t+1)} - \delta))$
11: **for** $k = 1$ to $K - 2$ **do**
12:     $\tau_k^{(t+1)} \leftarrow \max(\tau_{k-1}^{(t+1)} + \delta, \min(\tau_k^{(t+1)}, \tau_{k+1}^{(t+1)} - \delta))$
13: **end for**
14: $\tau_{K-1}^{(t+1)} \leftarrow \min(0.999, \max(\tau_{K-2}^{(t+1)} + \delta, \tau_{K-1}^{(t+1)}))$
15: **return** $\mathcal{T}^{(t+1)}$

---

The gradient-based optimization incorporates constraint projection to maintain the ordering requirements essential for meaningful sentiment categorization. The projection operator ensures that thresholds remain strictly ordered while staying within the feasible sentiment range, preventing numerical instabilities and maintaining semantic coherence.

---

**Algorithm 5** Iterative Optimization Framework

---

**Require:** Initial threshold set $\mathcal{T}^{(0)}$
**Require:** Regularization parameter $\lambda > 0$
**Require:** Learning rate $\eta \in (0, 1)$
**Require:** Convergence tolerance $\epsilon > 0$
**Require:** Maximum iterations $T_{\max}$
**Ensure:** Optimized threshold set $\mathcal{T}^*$
1: $t \leftarrow 0$
2: **repeat**
3:     $\{S_1^{(t)}, \ldots, S_K^{(t)}\} \leftarrow \text{STRATUMASSIGNMENT}(\mathcal{T}^{(t)}, \{y_i\}_{i=1}^N)$
4:     $\mathcal{L}^{(t)} \leftarrow \text{OBJECTIVEEVALUATION}(\{S_k^{(t)}\}, \mathcal{T}^{(t)}, \lambda, \Delta_{\text{target}})$
5:     $\mathcal{T}^{(t+1)} \leftarrow \text{GRADIENTPROJECTION}(\mathcal{T}^{(t)}, \eta, \delta)$
6:     $t \leftarrow t + 1$
7: **until** $\|\mathcal{T}^{(t)} - \mathcal{T}^{(t-1)}\|_2 < \epsilon$ **or** $t \geq T_{\max}$
8: $\mathcal{T}^* \leftarrow \mathcal{T}^{(t)}$
9: **return** $\mathcal{T}^*$

---

The iterative optimization framework orchestrates the convergence process through repeated application of the core subroutines. The convergence criterion balances computational efficiency with solution quality, terminating when threshold changes become negligible or when the maximum iteration limit is reached.

---

**Algorithm 6** Final Mapping Construction

---

**Require:** Optimal threshold set $\mathcal{T}^*$
**Require:** Original dataset $\{(x_i, y_i)\}_{i=1}^N$
**Require:** Category labels $\{z_1, z_2, \ldots, z_K\}$
**Ensure:** Mapping function $f^* : [-1, 1] \to \{z_1, \ldots, z_K\}$
**Ensure:** Enhanced dataset $\mathcal{D}' = \{(x_i, y_i, z_i)\}_{i=1}^N$
 1: **Apply Optimal Thresholds to Dataset**
 2: **for** $i = 1$ to $N$ **do**
 3:    **if** $y_i \leq \tau_0^*$ **then**
 4:       $z_i \leftarrow z_1$ {Assign to most negative category}
 5:    **else if** $y_i > \tau_{K-1}^*$ **then**
 6:       $z_i \leftarrow z_K$ {Assign to most positive category}
 7:    **else**
 8:       **for** $k = 2$ to $K - 1$ **do**
 9:          **if** $\tau_{k-1}^* < y_i \leq \tau_k^*$ **then**
10:            $z_i \leftarrow z_k$ {Assign to appropriate intermediate category}
11:            **break**
12:          **end if**
13:       **end for**
14:    **end if**
15: **end for**
16: **Construct Final Outputs**
17: Define $f^*(y) = z_k$ if $y \in I_k(\mathcal{T}^*)$ where intervals are defined by optimal thresholds
18: $\mathcal{D}' \leftarrow \{(x_i, y_i, z_i)\}_{i=1}^N$ {Enhanced dataset with discrete labels}
19: **return** $f^*, \mathcal{D}'$

---

The final mapping construction phase transforms the optimized threshold configuration into a practical classification system. This deterministic mapping preserves the semantic ordering of sentiment categories while providing the discrete labels required for classification tasks, resulting in an enhanced dataset suitable for multi-task learning frameworks.

A.16.3 ALGORITHM ANALYSIS

**Adaptivity.** The algorithm demonstrates superior adaptivity through its data-driven initialization and gradient-based optimization. Unlike fixed-threshold approaches, our method automatically adjusts to the specific characteristics of each dataset. The quantile-based initialization ensures that the algorithm starts from a reasonable configuration regardless of the underlying data distribution. The gradient optimization then fine-tunes these initial thresholds based on the actual variance structure within each stratum. This adaptivity is particularly crucial in sentiment analysis where different domains (financial, social media, product reviews) exhibit vastly different sentiment distributions.

**Balance Preservation.** The regularization term $\lambda \sum_{k=1}^{K-1} (\tau_k - \tau_{k-1} - \Delta_{\text{target}})^2$ serves as a balance-preserving mechanism that prevents the algorithm from creating severely skewed partitions. Without this term, the optimization might converge to solutions where some categories contain very few samples, leading to poor classification performance. The regularization parameter $\lambda$ provides explicit control over the trade-off between homogeneity and balance, allowing practitioners to tune the algorithm for their specific requirements.

**Optimality.** Under mild regularity conditions, the algorithm converges to a local minimum of the objective function. The convex nature of the variance term within each stratum, combined with the quadratic regularization term, ensures that the optimization landscape is well-behaved. While global optimality cannot be guaranteed due to the non-convex nature of the threshold-dependent stratum assignments, the careful initialization strategy typically leads to high-quality local optima that perform well in practice.

**Robustness.** The algorithm exhibits robustness against outliers and distributional shifts through multiple mechanisms. The variance-based objective naturally downweights the influence of extreme outliers within each stratum. The regularization term prevents overfitting to specific distributional artifacts in the training data. Additionally, the constrained optimization framework ensures that the

resulting thresholds remain within reasonable bounds, preventing pathological solutions that might work well on the training data but generalize poorly.

The overall computational complexity is $\mathcal{O}(T_{\max} \cdot K \cdot N)$ where $T_{\max}$ is the maximum number of iterations (typically 50-100), $K$ is the number of categories (typically 3-7), and $N$ is the dataset size. The space complexity is $\mathcal{O}(N+K)$, making the algorithm practical for large-scale applications. The parameter $\delta$ in the projection step is typically set to $10^{-6}$ to ensure numerical stability while maintaining strict inequality constraints.

## APPENDIX A.17: THE DUAL-WEIGHT MECHANISM ANALYSIS

### A.17.1 FORMAL PROBLEM FORMULATION

Consider a multi-task learning framework with tasks $\mathcal{T} = \{T_r, T_c\}$ representing regression and classification objectives, respectively. Let $\theta \in \mathbb{R}^d$ denote the shared model parameters, and $L_i^t : \mathbb{R}^d \to \mathbb{R}^+$ represent the loss function for task $i \in \{r, c\}$ at training step $t$. The fundamental challenge lies in determining optimal task weighting functions $\omega_i^t : \mathcal{S} \to \mathbb{R}^+$ that map the current training state $s^t \in \mathcal{S}$ to appropriate task weights, where $\mathcal{S}$ encompasses gradient information, batch characteristics, and historical performance.

Our dual-weight mechanism decomposes this weighting function into two components:

$$\omega_i^t(s^t) = \lambda_i^t(g^t) \cdot w_i^t(s^t), \tag{32}$$

where $\lambda_i^t(g^t)$ represents gradient-based normalization dependent on gradient state $g^t = \{\nabla_\theta L_r^t, \nabla_\theta L_c^t\}$, and $w_i^t(s^t)$ denotes learnable strategic weights dependent on the full training state.

### A.17.2 GRADIENT-BASED STABILIZATION: MATHEMATICAL DERIVATION

**Gradient Magnitude Problem.** In multi-task optimization, the combined gradient is:

$$\nabla_\theta L_{\mathrm{mtl}}^t = \sum_{i \in \{r,c\}} \omega_i^t \nabla_\theta L_i^t. \tag{33}$$

Without proper weighting, tasks with inherently larger gradient magnitudes dominate the optimization direction. Let $\|\nabla_\theta L_r^t\|_2 = g_r^t$ and $\|\nabla_\theta L_c^t\|_2 = g_c^t$ denote the L2 norms of task-specific gradients. The gradient magnitude imbalance ratio is:

$$\rho^t = \frac{\max(g_r^t, g_c^t)}{\min(g_r^t, g_c^t)}. \tag{34}$$

**Normalization Mechanism.** Our gradient-based weights implement adaptive normalization:

$$\lambda_r^t = \frac{g_c^t}{g_r^t + g_c^t}, \quad \lambda_c^t = \frac{g_r^t}{g_r^t + g_c^t}. \tag{35}$$

This formulation ensures $\lambda_r^t + \lambda_c^t = 1$ and provides inverse weighting where tasks with larger gradients receive smaller coefficients.

**Stability Guarantee.** The normalized combined gradient magnitude satisfies:

$$\|\lambda_r^t \nabla_\theta L_r^t + \lambda_c^t \nabla_\theta L_c^t\|_2 \leq \max(g_r^t, g_c^t), \tag{36}$$

with equality achieved when gradients are parallel. This bound prevents unbounded gradient growth regardless of individual task gradient magnitudes.

### A.17.3 WEIGHT LEARNING

**DWA Module Formulation.** The strategic weight generation follows:

$$\mathbf{h}^t = \mathrm{ReLU}(\mathbf{W}_1[\lambda_r^t L_r^t; \lambda_c^t L_{\mathrm{imb}}^t; \mathbf{f}(D^t)] + \mathbf{b}_1),$$
$$\mathbf{s}^t = \mathbf{W}_2 \mathbf{h}^t + \mathbf{b}_2, \tag{37}$$
$$[w_r^t, w_c^t] = \mathrm{Softmax}(\mathbf{s}^t),$$

where $\mathbf{f}(D^t)$ represents batch-specific features, $\mathbf{W}_1 \in \mathbb{R}^{h \times (2 + |\mathbf{f}(D^t)|)}$ and $\mathbf{W}_2 \in \mathbb{R}^{2 \times h}$ are learnable parameters.

**Expressivity Analysis.** The DWA module with ReLU activation can approximate any continuous function mapping batch characteristics to task weights, given sufficient hidden units. Specifically, for any continuous function $f : \mathcal{K} \to \Delta^1$ (where $\mathcal{K}$ is compact and $\Delta^1$ is the 1-simplex), there exists a DWA configuration achieving arbitrary approximation accuracy.

**Lyapunov Stability Analysis.** Define the combined loss as a potential function:

$$V^t(\theta) = \lambda_r^t w_r^t L_r^t(\theta) + \lambda_c^t w_c^t L_c^t(\theta). \tag{38}$$

**Proof Sketch.** The gradient-based normalization ensures bounded gradients at each step. The learnable weights converge due to the finite capacity of the DWA module and the use of standard optimization techniques. Combined with the convexity assumption, standard convergence results for stochastic gradient descent apply.

A.17.4 DWA'S ADVANTAGES

**Constant Weight Analysis.** Fixed weights $w_r, w_c$ lead to the optimization objective:

$$\min_\theta \mathbb{E}[w_r L_r^t(\theta) + w_c L_c^t(\theta)]. \tag{39}$$

This approach fails when $\mathbb{E}[g_r^t] \gg \mathbb{E}[g_c^t]$ or vice versa, leading to dominated optimization.

**Single Adaptive Weight Analysis.** Using only learnable weights without gradient normalization results in:

$$\min_\theta \mathbb{E}[w_r^t L_r^t(\theta) + w_c^t L_c^t(\theta)], \tag{40}$$

where $w_i^t$ adapt but cannot compensate for gradient magnitude imbalances within each step.

**Optimality Gap Analysis.** Let $\theta_{\text{dual}}^*$, $\theta_{\text{const}}^*$, and $\theta_{\text{single}}^*$ denote optimal solutions under dual-weight, constant-weight, and single adaptive weight schemes, respectively. Under mild regularity conditions:

$$V(\theta_{\text{dual}}^*) \leq V(\theta_{\text{single}}^*) \leq V(\theta_{\text{const}}^*), \tag{41}$$

where $V(\cdot)$ represents the true multi-task objective value.

## APPENDIX A.18: DWA FRAMEWORK SUMMARY

Traditional multi-task learning with constant weight can lead to suboptimal performance. Our proposed framework incorporates the DWA module that dynamically adjusts task weights based on their relative importance, gradients, and data characteristics during training. This batch-level dynamic adaptive loss addresses the limitations of constant-weight approaches by considering the varying difficulties and contributions of each task and data characteristics in each batch. The plug-and-play nature of the DWA module allows for flexible integration into various multi-task learning scenarios, making it applicable to a wide range of tasks.

## APPENDIX A.19: MULTI-TASK LOSS

Figure 7 demonstrates the superior convergence behavior of our proposed DWA framework compared to constant-weight approaches, where the dynamic adaptive loss (orange line) achieves faster initial convergence and maintains more stable optimization throughout training, while the weighted loss (blue line) exhibits slower but consistent improvement, ultimately validating the effectiveness of our adaptive weighting mechanism in balancing multi-task learning objectives.

Figure 8 illustrates the training dynamics under constant-weight multi-task learning, where the classification loss (cyan line) dominates the early training phases with significantly higher values compared to the regression loss (orange line), and the overall multi-task learning loss (blue line) follows the classification loss pattern, demonstrating the fundamental limitation of fixed weighting strategies in handling tasks with different convergence rates and loss magnitudes.

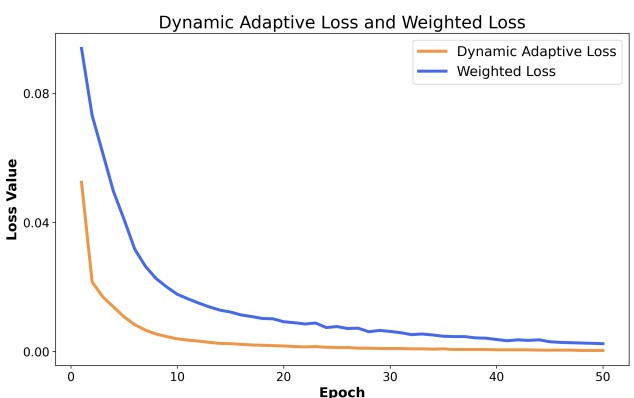

Figure 7: Comparison between constant-weight loss and DWA loss.

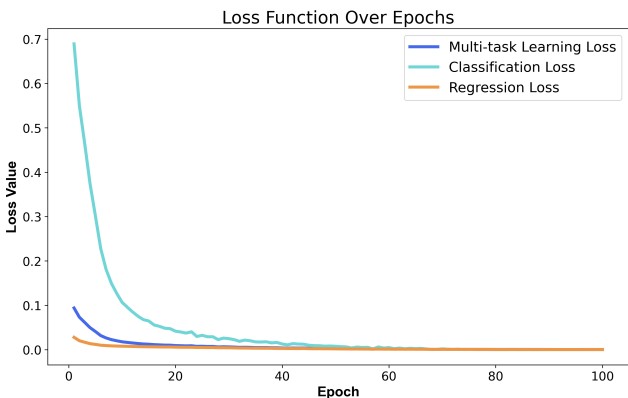

Figure 8: Constant-weight loss.

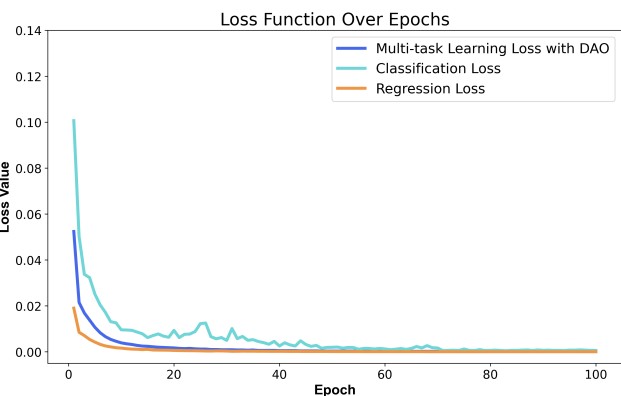

Figure 9: DWA loss.

Figure 9 presents the loss evolution with our DWA module, showing how the multi-task learning loss with DWA (blue line) effectively balances the contribution between classification loss $L_{imb}$ (cyan line) and regression loss (orange line), where the DWA mechanism successfully mitigates the dominance of classification loss observed in constant-weight approaches and achieves more harmonious multi-task optimization with improved convergence stability.