# OpenReview forum: "Dynamic Multi-Task Weight Adaptation for Efficient Sentiment Analysis Fine-Tuning on LLMs"
_ICLR.cc/2026/Conference — ICLR 2026 Conference Withdrawn Submission_

### Official Review · Reviewer_zzR1 · 2025-10-27

**Soundness:** 3
**Presentation:** 3
**Contribution:** 2
**Rating:** 4
**Confidence:** 4

**Summary:**

This paper investigates the imbalance problem in sentiment analysis for large language models (LLMs). Specifically, it first identifies the data imbalance issue in existing datasets and proposes an adaptive threshold selection optimization to address it. In addition, the paper introduces an auxiliary classification task to enhance the primary regression task. Furthermore, a Dynamic Weight Adaptor (DWA) is employed to adaptively balance the learning of regression and classification objectives. Extensive experiments demonstrate the effectiveness of the proposed method.

**Strengths:**

1. The imbalance problem is a long-standing challenge in the ML community, and studying it in the context of LLM fine-tuning remains important and relevant.
2. The proposed data-level and task-level rebalancing strategies appear to be technically sound and well-motivated.
3. Overall, the paper is clearly written and well-organized, making it easy to follow.

**Weaknesses:**

1. The proposed Dynamic Weight Adaptor (DWA) lacks novelty, and no comparisons are made with more advanced MTL approaches.
2. The experimental results appear relatively weak, with only marginal improvements over the baselines.
3. The employed models are not sufficiently large. Conducting experiments on larger models would be essential, as it remains unclear whether the imbalance problem persists at larger scales [1].
4. Some existing imbalance-handling methods require only simple hyperparameter tuning [2]; hence, a comparison with such approaches would strengthen the evaluation.
5. Experimental results should be reported as averages over multiple random seeds to ensure robustness and reproducibility.

Reference:

[1] PiKE: Adaptive Data Mixing for Multi-Task Learning Under Low Gradient Conflicts. ICLR 2025 Workshop.

[2] Escaping saddle points for effective generalization on class-imbalanced data. NeurIPS 2022.

**Questions:**

1. Could the authors clarify the exact roles and purposes of $\omega_r^t$ and $\omega_c^t$?
2. What do Method 1, Method 2, and Method 3 in Table 4 specifically refer to??

---

### Official Review · Reviewer_ujPn · 2025-10-28

**Soundness:** 2
**Presentation:** 2
**Contribution:** 1
**Rating:** 2
**Confidence:** 3

**Summary:**

This paper proposes a multi-task learning (MTL) framework to address the challenge of imbalanced data distributions in financial sentiment analysis, where neutral sentiments dominate. The authors augment a primary regression task with an auxiliary classification task. To support this, they introduce a Data-Aware Stratification (DAS) algorithm to create balanced category labels and a Dynamic Weight Adapter (DWA) module to automatically balance the losses from the two tasks. The framework is analyzed under both full fine-tuning and parameter-efficient fine-tuning using LoRA.

**Strengths:**

- The paper presents an extensive experimental analysis across multiple base models and includes a detailed ablation of LoRA integration, testing four different rank values.

- The paper is generally easy to read, and the motivation is intuitive. The authors clearly identify the problem of data imbalance and articulate the motivation behind their multi-task formulation.

**Weaknesses:**

- The paper introduces the challenge of imbalanced regression as the primary motivation for the work. However, the related work section omits any discussion of existing methods that address this problem (e.g., re-sampling, cost-sensitive regression, or imbalance-aware loss functions). The lack of comparison with these baselines makes it difficult to assess the contribution of the proposed MTL approach.

- The reported improvements (1.41% in accuracy and 12.36% relative improvement in MSE) correspond to very small absolute changes (e.g., from ~0.0220 to ~0.0188). The authors should clarify why this magnitude of improvement is practically meaningful in financial contexts. It would strengthen the paper to report performance across different sentiment regions (neutral vs. extreme) to demonstrate gains where imbalance is most severe.

- The paper states that only a training and testing split was provided at a ‘9:1 ratio’ (line 350). However, the results tables  report different epoch numbers for each run, which implies these results were chosen using the best performance on the test set. This raises concerns about evaluation protocol and reproducibility.

- The presentation of results could be significantly improved. Tables 2 and 3 are fragmented and inconsistently formatted, making it difficult to compare results across configurations. Consolidating the results into a single table (with rows configurations) and a new table for each dataset would greatly improve readability.

- The paper claims to use the Coefficient of Determination ($R^2$) for regression evaluation (line 342), but this metric does not appear in any of the main tables.

**Questions:**

Please see Weaknesses.

---

### Official Review · Reviewer_oLTm · 2025-11-02

**Soundness:** 3
**Presentation:** 3
**Contribution:** 2
**Rating:** 4
**Confidence:** 5

**Summary:**

This manuscript examines the suboptimal performance of large language models (LLMs) in the financial domain, which is caused by severe data distribution imbalance and inter-task difficulty discrepancies. It proposes a novel data-aware stratification (DAS) algorithm and a dynamic weight adapter (DWA) module. These two components are integrated into a multitask fine-tuning method, which has achieved satisfactory experimental results.

**Strengths:**

A.	The inter - task difficulty discrepancies and imbalanced data distribution are two significant issues in specific domain - related tasks. The motivation for addressing these problems is well - founded. Figures 2 and 3 attempt to illustrate the patterns reflected by these two issues in a relatively intuitive manner.
B.	The manuscript is supported by extensive experiments and detailed content in the appendix, which effectively substantiate its motivation.
C.	The model design strategy is relatively clear, which includes the Data-Aware Stratification (DAS) and the Dynamic Weight Adapter (DWA) module.
D.	The hyperparameters are presented in detail in the appendix.

**Weaknesses:**

A.	The issue of inter-task difficulty discrepancy is well-identified. However, Figure 2 lacks sufficient detail to convince readers that the experimental results are not coincidental. It is recommended to provide more details, such as the specific datasets used, the models employed, insights into the model representation patterns, additional experimental datasets, or references to other literature that have identified similar issues, to fully substantiate and support this finding.

B.	It is regrettable that the manuscript primarily employs small LLMs, such as RoBERTa, Qwen 0.5, and TinyLlama. The diversity of the models used is somewhat limited. Given the claim of plug-and-play capability, readers may expect to see the transferability of the method across a broader range of models. This limitation may be due to memory constraints or other practical considerations. It would be beneficial to address these limitations and provide justification for the choice of models.

C.	The manuscript lacks in-depth analysis regarding inter-task difficulty and sufficient support for the argument concerning inter - task difficulty discrepancy.

**Questions:**

Please refer to the points mentioned in the section titled "Weaknesses or Suggestions of the Manuscript."

---

### Official Review · Reviewer_mwSu · 2025-11-09

**Soundness:** 2
**Presentation:** 2
**Contribution:** 2
**Rating:** 2
**Confidence:** 4

**Summary:**

This paper proposes a multi-task learning framework for sentiment analysis on financial datasets, combining regression and classification objectives. The authors introduce two modules: (1) a Data-Aware Stratification (DAS) algorithm to rebalance data distribution via optimized thresholds, and (2) a Dynamic Weight Adapter (DWA) that adjusts task weights using gradient-based information and batch characteristics. Experiments on multiple datasets show modest improvements in MSE and accuracy compared to constant-weight baselines.

**Strengths:**

1 The overall writing and paper structure are clear and easy to follow.
2 The experiments show that the proposed approach is effective on certain models.

**Weaknesses:**

1 The methodological novelty is limited. The proposed modules largely extend existing ideas such as gradient-based weighting (e.g., GradNorm, uncertainty weighting) without offering substantial technical or conceptual innovation.

2 The experiments are constrained to a small set of models and do not consider more recent or stronger models (e.g., LLaMA, and GPT-based architectures), which limits the generalizability and impact of the results.

**Questions:**

NA

---

### Note · Authors · 2025-12-04

I have read and agree with the venue's withdrawal policy on behalf of myself and my co-authors.